# Efficient Causal Decision Making with One-sided Feedback

**Jianing Chu** *
Amazon
jianing.chu3@gmail.com

**Shu Yang & Wenbin Lu**
Department of Statistics
North Carolina State University
{syang24,wlu4}@ncsu.edu

**Pulak Ghosh**
Indian Institute of Management
pulak.ghosh@iimb.ac.in

## Abstract

We study a class of decision-making problems with one-sided feedback, where outcomes are only observable for specific actions. A typical example is bank loans, where the repayment status is known only if a loan is approved and remains undefined if rejected. In such scenarios, conventional approaches to causal decision evaluation and learning from observational data are not directly applicable. In this paper, we introduce a novel value function to evaluate decision rules that addresses the issue of undefined counterfactual outcomes. Without assuming no unmeasured confounders, we establish the identification of the value function using shadow variables. Furthermore, leveraging the semiparametric theory, we derive the efficiency bound for the proposed value function and develop efficient methods for decision evaluation and learning. Numerical experiments and a real-world data application demonstrate the empirical performance of our proposed methods.

## 1 Introduction

Binary decision-making problems are pervasive in the real world, encompassing domains such as bank loan approval (Pacchiano et al., 2021), job hiring (Raghavan et al., 2020), school admission (Baker & Hawn, 2022), and criminal recidivism prediction (Lakkaraju et al., 2017). Often, feedback in these scenarios is one-sided. Take bank loan approval as an example: a decision-maker is presented with covariates describing a loan applicant and decides whether to grant or deny the loan. If the loan is approved, feedback regarding the applicant's repayment is subsequently received. However, if the loan is denied, no further information is obtained. There are two main objectives in these decision-making processes: (1) evaluating a decision rule that aims to approve loans for applicants likely to repay while denying loans to those unlikely to do so, based on the expected outcomes it achieves; and (2) deriving an optimal decision rule that maximizes the expected outcome.

Decision-making with one-sided feedback can be viewed as a special contextual bandit problem with two actions, "approve" and "reject", where the outcome is observable exclusively when an individual is approved. Significant challenges arise due to the inherent heterogeneity between the approved and rejected groups—specifically, the conditional distribution of the outcome given the covariates may differ between these two groups. As a result, using an outcome model trained on approved samples to predict outcomes for the rejected group is generally unfeasible. To address model bias, one category of approaches uses exploration strategies to gather additional information from new samples, gradually reducing the bias over time (e.g. Jiang et al., 2021; Pacchiano et al., 2021). However, most existing works are restricted to binary outcomes and specific outcome models, lacking robustness to model misspecification and unable to generalize to numerical outcomes. Moreover, in real-world applications, exploration can be costly, risky, or even unethical, such as in

---

*This work was done prior to joining Amazon.

healthcare, finance, and education. This motivates us to develop practical approaches to decision evaluation and learning for different types of outcomes from observational data (Dudík et al., 2014; Kallus & Uehara, 2020; Athey & Wager, 2021; Chu et al., 2023a;b).

As mentioned above, disparities between approved and rejected groups often lead to variations in outcome measures due to unobserved differences in action selection, which also serve as predictors for the outcomes. This phenomenon violates a critical assumption in the causal inference literature for identifying and estimating the value function, known as the no unmeasured confounders (NUC) assumption (Imbens, 2004). This assumption, also referred to as strong ignorability (Rosenbaum & Rubin, 1983) or exogeneity (Imbens & Rubin, 2015), posits that actions are independent of potential outcomes given the covariates. Under this assumption, various approaches have been developed for estimating the value function, such as the inverse propensity weighting (IPW) method (Horvitz & Thompson, 1952) and the doubly robust (DR) method (Zhang et al., 2012; Dudík et al., 2014; Jiang & Li, 2016). The NUC assumption, however, can be often violated in many real-world scenarios. When the NUC assumption does not hold, the identifiability of the value function may be compromised, and existing estimators under this assumption may no longer be consistent for the value function.

To deal with such violations, the utilization of instrumental variables (IVs) emerges as a well-established strategy in the literature (Angrist et al., 1996; Hernán & Robins, 2006; Aronow & Carnegie, 2013; Wang & Tchetgen Tchetgen, 2018). An IV is defined as a pretreatment variable that is independent of all unmeasured confounders, and does not have a direct causal effect on the outcome other than through the action. However, it is acknowledged that identifying suitable IVs poses a considerable challenge, given the potential existence of numerous unmeasured confounders and the difficulty in eliminating the possibility of an IV's dependence on all of them. In contrast to IVs, we consider an alternative approach using a distinct type of variables known as shadow variables (SVs) (Wang et al., 2014; Shao & Wang, 2016; Miao et al., 2016; Li et al., 2024). SVs are independent of the action after conditioning on fully observed covariates and the outcome itself. Meanwhile, SVs are related to the outcome, potentially through unmeasured confounders. For example, in fairness-oriented employment, sensitive attributes such the age of candidates should be independent of the decision. However, these attributes may be related to the performance of candidates, thereby qualifying them as SVs. With the utilization of SVs, we show that the proposed value function is identifiable.

The contribution of this paper is multi-fold.

First, we propose a novel value function for decision-making with one-sided feedback. Without assuming the NUC condition, we consider a model that involves both outcomes and covariates for the action assignment mechanism. We provide identification for the proposed value function under this model by leveraging SVs.

Second, we derive the efficient influence function (EIF) and the semiparametric efficiency bound of the value function. Motivated by the EIF, we develop two different efficient estimators for the value function with binary and continuous outcomes, respectively. Our proposed estimation strategy does not require estimating the density when the outcome is continuous, thereby avoiding instability and distinguishing our methods from existing literature.

Third, we establish theoretical properties for the proposed estimators. We show the estimators are consistent and achieve the semiparametric efficiency bound under mild conditions of nuisance functions approximation.

Fourth, we propose a classification-based framework for learning the optimal decision rule, which allows us to leverage a wide range of existing classification tools tailored to different classes of decision rules. Through numerical experiments, we demonstrate that the proposed method significantly outperforms conventional decision learning methods.

## 2 RELATED WORK

**Contextual Bandits, Off-policy Evaluation and Learning** As formally described in Section 3, decision-making with one-sided feedback can be formulated as a special type of contextual bandits problem (Chu et al., 2011; Agrawal & Goyal, 2013; Zhou et al., 2020). There are a limited num-

ber of works focusing on one-sided feedback, with two notable related works in this setting. Jiang et al. (2021) considered binary outcomes and estimated outcome functions using generalized linear models, proposing an adaptive online learning approach that integrates uncertainty into outcome estimation. Pacchiano et al. (2021) studied the same problem setting with binary outcomes, approximating the outcome function using deep neural networks and proposing an online algorithm to train an optimistic decision-making model. However, their methods cannot be generalized to numerical outcomes and focus on the online learning setting. In contrast, the primary focus of our work is on decision evaluation and learning using observational data, commonly referred to as off-policy evaluation and learning in the context of contextual bandits. Off-policy methods have attracted significant interest, particularly in fields such as finance, medicine, and education, where experimentation and exploration can be risky, costly, or even unethical (Dudík et al., 2014; Kallus & Uehara, 2020; Athey & Wager, 2021; Chu et al., 2023a;b).

**Selective/Non-Random-Missing Labels** Although we study the problem under the contextual bandits setting, it is intrinsically related to the selective/non-random-missing labels problems in semi-supervised learning (Misra et al., 2016; Kleinberg et al., 2018; Sohn et al., 2020; Coston et al., 2021). In these problems, only a subset of instances receive labels, determined by the choices of decision-makers. This issue is further complicated by unmeasured confounders that influence both human decisions and the resulting outcomes. Lakkaraju et al. (2017) proposed a model evaluation method based on the assumption that the decisions in the historical dataset are made by different decision-makers with varying thresholds for their yes-no decisions. Sportisse et al. (2023) studied the problem in semi-supervised learning, adopting the assumption that the label-missing mechanism is independent of covariates given the label itself, implying that all covariates are SVs. Based on this assumption, they constructed consistent estimators for the loss function by modeling the label-missing mechanism. Hu et al. (2022) adopted the same assumption but proposed estimators without modeling the missing mechanism. The significant difference in our work is that we do not require all covariates to be SVs; instead, we allow the missing mechanism to depend on both the covariates and the outcome. More importantly, we develop the most efficient estimator by utilizing the semiparametric theory.

## 3 PRELIMINARIES

We consider a binary action $A \in \{0, 1\}$, where action 1 denotes "approve" and action 0 denotes "reject". Let $\mathbf{X} \in \mathcal{X} \subseteq \mathbb{R}^p$ denote a vector of covariates, and $Y \in \mathbb{R}$ denote the observed outcome of interest. We assume larger values of $Y$ are preferred by convention. We study the problem under the counterfactual potential-outcome framework (Rubin, 2005). The potential outcomes $Y(a)$, $a = 0, 1$, which are the outcomes that would be observed if a subject received action $a = 0$ or $a = 1$, both are well-defined in conventional decision-making problems. Under the Stable Unit Treatment Value Assumption (SUTVA) (Rubin, 2005), we have $Y = AY(1) + (1 - A)Y(0)$. However, under the one-sided feedback setting, only $Y(1)$ is defined, and the outcome $Y$ is only observed if an individual is approved ($A = 1$). In this case, the observed outcome is always $Y = Y(1)$. The observed data are then $\{\mathbf{O}_i = (Y_i A_i, A_i, \mathbf{X}_i), i = 1, \ldots, n\}$ and we assume they are independent and identically distributed.

A decision rule $\pi : \mathcal{X} \to [0, 1]$ is a map from covariates to a probability, so that a decision maker, when presented with covariates $\mathbf{X}$, will select action 1 with probability $\pi(\mathbf{X})$. In conventional decision-making, where potential outcomes are defined for both actions, implementing a decision rule $\pi$ in a population would yield the population mean outcome, commonly referred to as the value function, defined as follows:

$$V(\pi) = \mathbb{E}\left[Y(1)\pi(\mathbf{X}) + Y(0)\{1 - \pi(\mathbf{X})\}\right]. \tag{1}$$

Under the one-sided feedback setting, since $Y(0)$ is not defined, we can no longer use the definition of value function in (1). We define a new value function as

$$V_1(\pi) = \mathbb{E}\{Y(1)\pi(\mathbf{X})\}. \tag{2}$$

The interpretation of $V_1(\pi)$ is straightforward. Consider a practical example of bank loans and a deterministic decision rule $\pi$ (where $\pi(\mathbf{X})$ can only take on values 0 or 1). Let $Y(1)$ denote the money earned by the bank if a loan is approved. For an applicant with covariates $\mathbf{X}$, if $\pi(\mathbf{X}) = 1$,

indicating loan approval, then $Y(1)\pi(\mathbf{X}) = Y(1)$ represents the potential financial outcome for the bank. On the other hand, if $\pi(\mathbf{X}) = 0$, indicating loan rejection, the bank neither earns nor loses any money. Therefore, the newly defined value function $V_1(\pi)$ quantifies the expected monetary outcome for the bank when implementing decision rule $\pi$ for loan approvals. We define the optimal decision rule as the one that maximizes the defined value function: $\pi^* = \arg\max_{\pi \in \Pi} V_1(\pi)$. Our first goal is to evaluate a given decision rule $\pi$ by estimating $V_1(\pi)$ using the historical data $\{\mathbf{O}_i = (Y_i A_i, A_i, \mathbf{X}_i), i = 1, \ldots, n\}$. Our second goal is to learn the optimal decision rule $\pi^*$.

## 4    IDENTIFICATION, EIF, AND EFFICIENCY BOUND

In this section, we provide the identification of the value function $V_1(\pi)$, and establish the corresponding EIF and efficiency bound under the semiparametric theory.

### 4.1    IDENTIFICATION

Without assuming the NUC condition that $Y(1) \perp\!\!\!\perp A \mid \mathbf{X}$, we consider a general action assignment mechanism that depends not only on covariates but also on the potential outcome:

$$\varphi(\mathbf{x}, y) \equiv \mathbb{P}\{A = 1 \mid \mathbf{X} = \mathbf{x}, Y(1) = y\},$$

and we assume $0 < \varphi(\mathbf{x}, y) < 1$. Let $f(\mathbf{x})$ denote the marginal density of $\mathbf{X}$, and let $f(y \mid \mathbf{x}, 1)$ denote the conditional density of $Y(1)$ given $\mathbf{X} = \mathbf{x}$ and $A = 1$. Let $w(\mathbf{x}) \equiv \mathbb{P}(A = 1 \mid \mathbf{X} = \mathbf{x})$. We can show that the value function $V_1(\pi)$ has the following representation (details are given in Appendix A.1) :

$$V_1(\pi) = \mathbb{E}\{Y(1)\pi(\mathbf{X})\} = \int f(\mathbf{x})w(\mathbf{x})\left\{\int y\frac{f(y \mid \mathbf{x}, 1)}{\varphi(\mathbf{x}, y)}dy\right\}\pi(\mathbf{x})d\mathbf{x}. \tag{3}$$

Therefore, we can identify $V_1(\pi)$ through identifying $f(\mathbf{x})$, $w(\mathbf{x})$, $f(y \mid \mathbf{x}, 1)$, and $\varphi(\mathbf{x}, y)$. The likelihood function for a single observation is

$$f(\mathbf{x})w(\mathbf{x})^a\{1 - w(\mathbf{x})\}^{1-a}f(y \mid \mathbf{x}, 1)^a.$$

Thus, $f(\mathbf{x})$, $w(\mathbf{x})$, and $f(y \mid \mathbf{x}, 1)$ can be identified from the observed data distribution. However, as noted in the literature (e.g. Wang et al., 2014; Miao et al., 2016), $\varphi(\mathbf{x}, y)$ is not identifiable without further assumptions.

We assume that covariates $\mathbf{X}$ can be partitioned into two subsets of variables $\mathbf{U}$ and $\mathbf{Z}$, i.e. $\mathbf{X} = (\mathbf{U}^T, \mathbf{Z}^T)^T$. $\mathbf{U}$ and $\mathbf{Z}$ are variables satisfying the following assumptions.

**Assumption 4.1** $\mathbf{Z} \perp\!\!\!\perp A \mid \mathbf{U}, Y(1)$ *and* $\mathbf{Z} \not\!\perp\!\!\!\perp Y(1) \mid \mathbf{U}$.

**Assumption 4.2** *For any function* $h(Y(1), \mathbf{U})$, $\mathbb{E}\{h(Y(1), \mathbf{U}) \mid \mathbf{X}, A = 1\} = 0$ *implies* $h(Y(1), \mathbf{U}) = 0$ *almost surely.*

Assumption 4.1 indicates $\mathbf{Z}$ are SVs and $\varphi(\mathbf{x}, y) = \mathbb{P}\{A = 1 \mid \mathbf{X} = \mathbf{x}, Y(1) = y\} = \mathbb{P}\{A = 1 \mid \mathbf{U} = \mathbf{u}, Y(1) = y\} = \varphi(\mathbf{u}, y)$. For example, in fairness-oriented employment, sensitive attributes such as the age of candidates should be unrelated to the action assignment. If these attributes correlate with the performance of candidates, they can be considered SVs. SVs can be selected based on expert prior knowledge, or alternatively, representations that serve the role of shadow variables can be generated directly from observed covariates without the need for prior knowledge (Li et al., 2024). Assumption 4.2 is known as the conditional completeness assumption, which is widely used in identification problems (Newey & Powell, 2003; Miao et al., 2015; Yang et al., 2019). This condition guarantees the uniqueness of $\varphi(\mathbf{u}, y)$. When both $Y(1)$ and $\mathbf{Z}$ are categorical variables with $l$ and $m$ levels, respectively, Assumption 4.2 holds if $l < m$. When $Y(1)$ is continuous, Assumption 4.2 holds when $f(y \mid \mathbf{x}, 1)$ follows some common distributions, such as exponential families.

**Theorem 4.3** *Under Assumptions 4.1 and 4.2,* $f(\mathbf{x})$, $w(\mathbf{x})$, $f(y \mid \mathbf{x}, 1)$, *and* $\varphi(\mathbf{u}, y)$ *are identifiable, and thus* $V_1(\pi)$ *is identified by*

$$V_1(\pi) = \int f(\mathbf{x})w(\mathbf{x})\left\{\int y\frac{f(y \mid \mathbf{x}, 1)}{\varphi(\mathbf{u}, y)}dy\right\}\pi(\mathbf{x})d\mathbf{x}. \tag{4}$$

## 4.2 EIF AND EFFICIENCY BOUND

The identification (4) motivates a rich class of estimators for the value function. However, to guide the construction of more principled estimators, we derive the EIF and the efficiency bound for the value function using the semiparemetric theory (Bickel et al., 1993; Tsiatis, 2006) in this section. Semiparametric models are sets of probability distributions that indexed by both finite-dimensional parametric and infinite-dimensional nonparametric components. The semiparametric efficiency bound is defined as the supremum of the Cramer-Rao lower bounds for all parametric submodels. The EIF is the influence function of a semiparametric regular and asymptotically linear estimator that achieves the semiparametric efficiency bound. We assume a general model for the action assignment mechanism, denoted as $\varphi(\mathbf{u}, y; \eta)$, which is represented by a parameter $\eta$. Consider the Hilbert space $\mathcal{T}$ of all measurable functions of the observed data with mean zero and finite variance, equipped with covariance inner product $\langle h_1, h_2 \rangle = \mathbb{E}\{h_1(\cdot)^T h_2(\cdot)\}$, where $h_1, h_2 \in \mathcal{T}$. We first derive the nuisance tangent space and its orthogonal complement, where the nuisance tangent space is defined as the mean squared closure of all parametric submodel nuisance tangent spaces (Bickel et al., 1993; Tsiatis, 2006). For the ease of exposition, we simplify $\varphi(\mathbf{U}, Y(1); \eta)$ as $\varphi(\eta)$ and $\partial\varphi(\mathbf{U}, Y(1); \eta)/\partial\eta$ as $\dot{\varphi}(\eta)$.

**Theorem 4.4** *The Hilbert space $\mathcal{T}$ can be decomposed as*

$$\mathcal{T} = \Lambda_1 \oplus \Lambda_2 \oplus \Lambda_\perp,$$

*where*

$$\Lambda_1 = [h_1(\mathbf{X}) : \mathbb{E}\{h_1(\mathbf{X}) = 0\}],$$

$$\Lambda_2 = \left[ A h_2(\mathbf{X}, Y(1)) + \frac{w(\mathbf{X}) - A}{1 - w(\mathbf{X})} \mathbb{E}\{h_2(\mathbf{X}, Y(1)) \mid \mathbf{X}\} : \mathbb{E}\{h_2(\mathbf{X}, Y(1)) \mid \mathbf{X}, A = 1\} = 0 \right],$$

$$\Lambda_\perp = \left\{ \frac{\varphi(\eta) - A}{\varphi(\eta)} g(\mathbf{X}) \right\},$$

*$g(\mathbf{X})$ is a function with the same dimension as $\eta$, and the notation $\oplus$ denotes the direct sum of two spaces that are orthogonal to each other.*

Based on Theorem 4.4, the EIF for $V_1(\pi)$ has the following form

$$\phi_{\text{eff}} = \underbrace{h_1^*(\mathbf{X})}_{\in \Lambda_1} + \underbrace{A h_2^*(\mathbf{X}) + \frac{w(\mathbf{X}) - A}{1 - w(\mathbf{X})} \mathbb{E}\{h_2^*(\mathbf{X}, Y(1)) \mid \mathbf{X}\}}_{\in \Lambda_2} + \underbrace{\boldsymbol{D}^T S_{\eta,\text{eff}}}_{\in \Lambda_\perp},$$

where $\mathbb{E}\{h_1^*(\mathbf{X})\} = 0, \mathbb{E}\{h_2^*(\mathbf{X}, Y(1)) \mid \mathbf{X}, A = 1\} = 0$, $S_{\eta,\text{eff}}$ is the efficient score for $\eta$, and $\boldsymbol{D}$ is a vector with the same dimension as $\eta$. The efficient score $S_{\eta,\text{eff}}$ can be obtained by projecting the score function of $\eta$ onto $\Lambda_\perp$, as stated in the following theorem.

**Theorem 4.5** *Under Assumptions 4.1 and 4.2, the efficient score for $\eta$ is*

$$S_{\eta,\text{eff}} = \frac{\varphi(\eta) - A}{\varphi(\eta)} \frac{\mathbb{E}\left\{ \frac{\dot{\varphi}(\eta)}{\varphi(\eta)^2} \mid \mathbf{X}, A = 1 \right\}}{\mathbb{E}\left\{ \frac{\varphi(\eta) - 1}{\varphi(\eta)^2} \mid \mathbf{X}, A = 1 \right\}}.$$

By projecting the value function identification (4) onto $\Lambda_1, \Lambda_2$, and $\Lambda_\perp$, we can derive $h_1^*(\mathbf{X})$, $h_2^*(\mathbf{X})$, and $\boldsymbol{D}$. The EIF and semiparametric efficiency bound for the value function are given in the following theorem.

**Theorem 4.6** *Under Assumptions 4.1 and 4.2, the EIF for $V_1(\pi)$ is*

$$\phi_{\text{eff}}(\pi) = \pi(\mathbf{X}) \left[ \frac{A}{\varphi(\eta)} Y + \left\{ 1 - \frac{A}{\varphi(\eta)} \right\} \frac{\mathbb{E}\left\{ \frac{1 - \varphi(\eta)}{\varphi(\eta)^2} Y \mid \mathbf{X}, A = 1 \right\}}{\mathbb{E}\left\{ \frac{1 - \varphi(\eta)}{\varphi(\eta)^2} \mid \mathbf{X}, A = 1 \right\}} \right] - V_1(\pi) + \boldsymbol{D}^T S_{\eta,\text{eff}}, \quad (5)$$

*where $\boldsymbol{D} = \{\text{Var}(S_{\eta,\text{eff}})\}^{-1} \left( \mathbb{E}\left[ \pi(\mathbf{X}) \frac{\mathbb{E}\left\{ \frac{1 - \varphi(\eta)}{\varphi(\eta)^2} Y \mid \mathbf{X}, A = 1 \right\}}{\mathbb{E}\left\{ \frac{1 - \varphi(\eta)}{\varphi(\eta)^2} \mid \mathbf{X}, A = 1 \right\}} \frac{\dot{\varphi}(\eta)}{\varphi(\eta)} \right] - \mathbb{E}\left[ \pi(\mathbf{X}) \mathbb{E}\left\{ \frac{\dot{\varphi}(\eta)}{\varphi(\eta)^2} Y \mid \mathbf{X}, A = 1 \right\} \right] \right).$*

*The semiparametric efficiency bound for $V_1(\pi)$ is $\Upsilon(\pi) = \mathbb{E}\{\phi_{\text{eff}}^2(\pi)\}$.*

# 5 EFFICIENT DECISION EVALUATION AND LEARNING

## 5.1 EFFICIENT VALUE ESTIMATION

Based on the EIF (5), since $\boldsymbol{D}$ is a fixed vector and $S_{\eta,\mathrm{eff}}$ is a score function with mean zero, we propose the following estimator for $V_1(\pi)$:

$$
\widehat{V}_1(\pi) = \mathbb{P}_n\left(\pi(\mathbf{x})\left[\frac{a}{\varphi(\widehat{\eta})}y + \left\{1 - \frac{a}{\varphi(\widehat{\eta})}\right\}\frac{\widehat{\mathbb{E}}\left\{\frac{1-\varphi(\eta)}{\varphi(\eta)^2}Y \mid \mathbf{x}, 1\right\}}{\widehat{\mathbb{E}}\left\{\frac{1-\varphi(\eta)}{\varphi(\eta)^2} \mid \mathbf{x}, 1\right\}}\right]\right),
\tag{6}
$$

where $\mathbb{P}_n[h(\mathbf{x})] = \frac{1}{n}\sum_{i=1}^{n}h(\mathbf{x}_i)$ for any given function $h(\mathbf{x})$, and quantities marked with hats are estimates of their unmarked counterparts. To obtain the value estimator, we first need to estimate $\eta$ and two conditional expectations $\mathbb{E}\left\{\frac{1-\varphi(\eta)}{\varphi(\eta)^2}Y \mid \mathbf{x}, 1\right\}$ and $\mathbb{E}\left\{\frac{1-\varphi(\eta)}{\varphi(\eta)^2} \mid \mathbf{x}, 1\right\}$. A general semiparametric estimator for $\eta$ can be obtained by solving the following equation:

$$
\mathbb{P}_n\left[\frac{\varphi(\mathbf{u}, y; \eta) - a}{\varphi(\mathbf{u}, y; \eta)}g(\mathbf{x}; \eta)\right] = 0,
\tag{7}
$$

where $g(\mathbf{x}; \eta)$ is a calibration function with the same dimension as $\eta$. Although this estimator achieves consistency and asymptotic normality under certain regularity conditions, its efficiency is not guaranteed. To ensure minimum estimation variability introduced by $\widehat{\eta}$, we need to derive the efficient estimator of $\eta$, denoted as $\widehat{\eta}_{\mathrm{eff}}$. This estimator can be obtained by solving the estimation equation based on the efficient score $S_{\eta,\mathrm{eff}}$ given in Theorem 4.5,

$$
\mathbb{P}_n\left[\frac{\varphi(\eta) - a}{\varphi(\eta)}\frac{\mathbb{E}\left\{\frac{\dot{\varphi}(\eta)}{\varphi(\eta)^2} \mid \mathbf{x}, 1\right\}}{\mathbb{E}\left\{\frac{\varphi(\eta)-1}{\varphi(\eta)^2} \mid \mathbf{x}, 1\right\}}\right] = 0.
\tag{8}
$$

However, the closed forms of the two conditional expectations in (8) are unknown and need to be approximated. We consider the following two scenarios.

**Scenario I:** When the outcome $Y$ is binary, say $Y \in \{0, 1\}$, we can specify a model for $\mathbb{P}(Y = 1 \mid \mathbf{X}, A = 1)$ and we denote its estimator as $\widehat{\mathbb{P}}(Y = 1 \mid \mathbf{X}, A = 1)$. The conditional expectations in (8) can be estimated by $\widehat{\mathbb{E}}\left\{\frac{\dot{\varphi}(\eta)}{\varphi(\eta)^2} \mid \mathbf{X}, A = 1\right\} = \frac{1}{\varphi(\mathbf{U},1;\eta)^2}\frac{\partial\varphi(\mathbf{U},1;\eta)}{\partial\eta}\widehat{\mathbb{P}}(Y = 1 \mid \mathbf{X}, A = 1) + \frac{1}{\varphi(\mathbf{U},0;\eta)^2}\frac{\partial\varphi(\mathbf{U},0;\eta)}{\partial\eta}\{1 - \widehat{\mathbb{P}}(Y = 1 \mid \mathbf{X}, A = 1)\}$, and $\widehat{\mathbb{E}}\left\{\frac{\varphi(\eta)-1}{\varphi(\eta)^2} \mid \mathbf{X}, A = 1\right\} = \frac{\varphi(\mathbf{U},1;\eta)-1}{\varphi(\mathbf{U},1;\eta)^2}\widehat{\mathbb{P}}(Y = 1 \mid \mathbf{X}, A = 1) + \frac{\varphi(\mathbf{U},0;\eta)-1}{\varphi(\mathbf{U},0;\eta)^2}\{1 - \widehat{\mathbb{P}}(Y = 1 \mid \mathbf{X}, A = 1)\}$. Thus we can get the efficient estimator $\widehat{\eta}_{\mathrm{eff}}$ by solving (8). Next, the conditional expectations in (6) can be estimated by $\widehat{\mathbb{E}}\left\{\frac{1-\varphi(\eta)}{\varphi(\eta)^2}Y \mid \mathbf{X}, A = 1\right\} = \frac{1-\varphi(\mathbf{U},1;\widehat{\eta}_{\mathrm{eff}})}{\varphi(\mathbf{U},1;\widehat{\eta}_{\mathrm{eff}})^2}\widehat{\mathbb{P}}(Y = 1 \mid \mathbf{X}, A = 1)$, and $\widehat{\mathbb{E}}\left\{\frac{1-\varphi(\eta)}{\varphi(\eta)^2} \mid \mathbf{X}, A = 1\right\} = \frac{1-\varphi(\mathbf{U},1;\widehat{\eta}_{\mathrm{eff}})}{\varphi(\mathbf{U},1;\widehat{\eta}_{\mathrm{eff}})^2}\widehat{\mathbb{P}}(Y = 1 \mid \mathbf{X}, A = 1) + \frac{1-\varphi(\mathbf{U},0;\widehat{\eta}_{\mathrm{eff}})}{\varphi(\mathbf{U},0;\widehat{\eta}_{\mathrm{eff}})^2}\{1 - \widehat{\mathbb{P}}(Y = 1 \mid \mathbf{X}, A = 1)\}$. By plugging the estimators $\widehat{\eta}_{\mathrm{eff}}$, $\widehat{\mathbb{E}}\left\{\frac{1-\varphi(\eta)}{\varphi(\eta)^2}Y \mid \mathbf{X}, A = 1\right\}$, and $\widehat{\mathbb{E}}\left\{\frac{1-\varphi(\eta)}{\varphi(\eta)^2} \mid \mathbf{X}, A = 1\right\}$ into (6), we obtain the value estimator and denote it as $\widehat{V}_{\mathrm{eff}}(\pi)$.

**Scenario II:** When the outcome $Y$ is continuous, one can still first model the conditional density $f(Y \mid \mathbf{X}, A = 1)$. However, the density estimation often requires large sample sizes and complex algorithms to achieve accurate estimates. This can be computationally intensive and prone to high variance, particularly in high-dimensional spaces. Instead, we propose a two-step estimation strategy. In step 1, we find a root-$n$ consistent estimator $\widehat{\eta}^{(1)}$. For example, we can choose a simple calibration function $g(\mathbf{x}; \eta)$ and solve the equation (7). In step 2, we construct pseudo-outcomes $\frac{\dot{\varphi}(\widehat{\eta}^{(1)})}{\varphi^2(\widehat{\eta}^{(1)})}$ and $\frac{\varphi(\widehat{\eta}^{(1)})-1}{\varphi^2(\widehat{\eta}^{(1)})}$ and the estimators of the conditional expectations, $\widehat{\mathbb{E}}\left\{\frac{\dot{\varphi}(\eta)}{\varphi(\eta)^2} \mid \mathbf{X}, A = 1\right\}$ and $\widehat{\mathbb{E}}\left\{\frac{\varphi(\eta)-1}{\varphi(\eta)^2} \mid \mathbf{X}, A = 1\right\}$ can then be obtained using regression with these pseudo-outcomes. Thus we can get the efficient estimator $\widehat{\eta}_{\mathrm{eff}}$ by solving (8). Similarly, to estimate the conditional expectations in (6), we can construct pseudo-outcomes $\frac{1-\varphi(\widehat{\eta}_{\mathrm{eff}})}{\varphi(\widehat{\eta}_{\mathrm{eff}})^2}Y$ and $\frac{1-\varphi(\widehat{\eta}_{\mathrm{eff}})}{\varphi(\widehat{\eta}_{\mathrm{eff}})^2}$. The

estimators $\widehat{\mathbb{E}}\left\{\frac{1-\varphi(\eta)}{\varphi(\eta)^2}Y \mid \mathbf{X}, A=1\right\}$, and $\widehat{\mathbb{E}}\left\{\frac{1-\varphi(\eta)}{\varphi(\eta)^2} \mid \mathbf{X}, A=1\right\}$ can be obtained using regression with these pseudo-outcomes. By plugging the estimators $\widehat{\eta}_{\text{eff}}$, $\widehat{\mathbb{E}}\left\{\frac{1-\varphi(\eta)}{\varphi(\eta)^2}Y \mid \mathbf{X}, A=1\right\}$, and $\widehat{\mathbb{E}}\left\{\frac{1-\varphi(\eta)}{\varphi(\eta)^2} \mid \mathbf{X}, A=1\right\}$ into (6), we obtain the value estimator and denote it as $\widehat{V}_{\text{eff}}(\pi)$.

We now establish the theoretical results for the proposed value estimator. We first make the following assumptions for the nuisance functions and their approximations.

**Assumption 5.1** *For all* $\mathbf{x} \in \mathcal{X}$, *(i)* $\{|k_1(\mathbf{x})|, |\widehat{k}_1(\mathbf{x})|\} > 0$, *where* $k_1(\mathbf{x}) = \widehat{\mathbb{E}}\left\{\frac{\varphi(\eta)-1}{\varphi(\eta)^2} \mid \mathbf{x}, 1\right\}$; *(ii) for any* $k_2(\mathbf{x}) \in \left\{\mathbb{E}\left\{\frac{\dot{\varphi}(\eta)}{\varphi(\eta)^2} \mid \mathbf{x}, 1\right\}, \mathbb{E}\left\{\frac{1-\varphi(\eta)}{\varphi(\eta)^2}Y \mid \mathbf{x}, 1\right\}\right\}$, $\{|k_2(\mathbf{x})|, |\widehat{k}_2(\mathbf{x})|\} < \infty$. *(iii) for any* $k_3(\mathbf{x}) \in \left\{\mathbb{E}\left\{\frac{\varphi(\eta)-1}{\varphi(\eta)^2} \mid \mathbf{x}, 1\right\}, \mathbb{E}\left\{\frac{1-\varphi(\eta)}{\varphi(\eta)^2}Y \mid \mathbf{x}, 1\right\}, \mathbb{E}\left\{\frac{\dot{\varphi}(\eta)}{\varphi(\eta)^2} \mid \mathbf{x}, 1\right\}\right\}$, $\widehat{k}_3(\mathbf{x}) \xrightarrow{p} k_3(\mathbf{x})$.

Assumption 5.1 (i) and (ii) require that the conditional expectations and their estimations are bounded. Assumption 5.1 (iii) requires that the conditional expectations are consistently estimated. In the case of a binary outcome, the estimation of $\mathbb{P}(Y=1 \mid \mathbf{X}, A=1)$ is required to be consistent. For continuous outcomes, given the root-$n$ consistency of $\widehat{\eta}^{(1)}$, we only require that the regression with constructed pseudo-outcomes is consistent. This can be achieved by various machine and deep learning models (e.g. Kennedy, 2016; Farrell et al., 2021).

**Theorem 5.2** *Under Assumptions 4.1, 4.2, and 5.1 (i) (ii),* $\widehat{V}_{\text{eff}}(\pi)$ *is a consistent estimator for* $V_1(\pi)$. *Additionally, if Assumption 5.1 (iii) holds,* $\widehat{V}_{\text{eff}}(\pi)$ *achieves the semiparametric efficiency bound* $\Upsilon(\pi)$.

## 5.2 FROM EFFICIENT DECISION EVALUATION TO LEARNING

In this section, we consider a deterministic decision rule class $\Pi$ and propose a method based on the efficient estimator $\widehat{V}_{\text{eff}}(\pi)$ to learn the optimal decision rule, $\pi^* = \arg\max_{\pi \in \Pi} V_1(\pi)$. A natural estimator for the optimal decision rule $\pi^*$ would be $\widehat{\pi} = \arg\max_{\pi \in \Pi} \widehat{V}_{\text{eff}}(\pi)$. However, this direct search poses a significant challenge as it typically involves non-convex and non-smooth optimization problems and can be computationally expensive. We have the following proposition to transform it into a weighted classification problem.

**Proposition 5.3** *Maximizing the value estimator* $\widehat{V}_{\text{eff}}(\pi)$ *is equivalent to a weighted classification problem of minimizing the following loss function over* $\pi \in \Pi$,

$$n^{-1}\sum_{i=1}^{n} \mathbb{I}\{\mathbb{I}\{\widehat{\psi}(\mathbf{x}_i, y_i, a_i) > 0\} \neq \pi(\mathbf{x}_i)\}|\widehat{\psi}(\mathbf{x}_i, y_i, a_i)|, \tag{9}$$

*where* $\widehat{\psi}(\mathbf{x}_i, y_i, a_i) = \frac{a_i}{\varphi_i(\widehat{\eta}_{\text{eff}})}y_i + \left\{1 - \frac{a_i}{\varphi_i(\widehat{\eta}_{\text{eff}})}\right\}\frac{\widehat{\mathbb{E}}\left\{\frac{1-\varphi(\eta)}{\varphi(\eta)^2}Y \mid \mathbf{x}_i, 1\right\}}{\widehat{\mathbb{E}}\left\{\frac{1-\varphi(\eta)}{\varphi(\eta)^2} \mid \mathbf{x}_i, 1\right\}}$, *for* $1 \leq i \leq n$.

With Proposition 5.3, we have transformed the optimal decision rule learning into a weighted classification problem (9) where for subject $i$ with features $\mathbf{x}_i$, the true label is $\mathbb{I}\{\widehat{\psi}(\mathbf{x}_i, y_i, a_i) > 0\}$ and the sample weight is $|\widehat{\psi}(\mathbf{x}_i, y_i, a_i)|$. The choice of classification approach dictates the restricted class $\Pi$. Compared to a direct search, a classification-based optimizer facilitates handling more complex functional classes and allows for the use of off-the-shelf machine learning and deep learning software packages.

## 6 EXPERIMENTS

We have carried out extensive simulation studies and a real data application to evaluate the performance of the proposed methods.

## 6.1 SYNTHETIC SCENARIOS

We compare the proposed method with three alternative methods. One consistent but not efficient estimator for $\eta$ is the solution to the estimation equation (7) with a simple choice $g(\mathbf{x}; \eta)$. We denote this estimator as $\widehat{\eta}_{\text{naive}}$. The first estimator for the value function is the IPW estimator with $\widehat{\eta}_{\text{naive}}$: $\widehat{V}_{\text{IPW-naive}}(\pi) = \mathbb{P}_n \left[ \frac{a}{\varphi(\widehat{\eta}_{\text{naive}})} y \pi(\mathbf{x}) \right]$. The second estimator is also an IPW estimator but with $\widehat{\eta}_{\text{eff}}$: $\widehat{V}_{\text{IPW-eff}}(\pi) = \mathbb{P}_n \left[ \frac{a}{\varphi(\widehat{\eta}_{\text{eff}})} y \pi(\mathbf{x}) \right]$. The third estimator is the DR estimator (Zhang et al., 2012; Dudík et al., 2014): $\widehat{V}_{\text{DR}}(\pi) = \mathbb{P}_n \left( \pi(\mathbf{x}) \left[ \frac{a}{\widehat{w}(\mathbf{x})} \left\{ y - \widehat{\mathbb{E}}(y \mid \mathbf{x}) \right\} + \widehat{\mathbb{E}}(y \mid \mathbf{x}) \right] \right)$.

**Decision Evaluation:** We first generate covariates $\mathbf{X} = (X_1, X_2, X_3)^T \sim N((1, -1, 0)^T, \Sigma)$, where $\Sigma = \begin{pmatrix} 1 & -0.25 & -0.25 \\ -0.25 & 1 & -0.25 \\ -0.25 & -0.25 & 1 \end{pmatrix}$. We consider two types of potential outcome, continuous and binary.

Case 1: The potential outcome $Y(1)$ is generated by $Y(1) = 8X_1 - 4X_1^2 - 4X_2 + 4X_3^2 + \epsilon$, where $\epsilon$ is generated from a normal distribution with mean 0 and standard deviation 0.5. The action $A$ is generated from $A \sim \text{Bernoulli}\{\varphi(\mathbf{X}, Y(1))\}$, and $\text{logit}\{\varphi(\mathbf{X}, Y(1))\} = 1/[1 + \exp\{0.5 - X_1 - X_2 - 0.1Y(1)\}]$. Thus, $X_3$ is the shadow variable. We construct three different evaluation decision rules as mixtures of a deterministic decision rule $\pi_d(\mathbf{X}) = \mathbb{I}(2X_1 - X_1^2 - X_2 + X_3^2 > 0)$ and the uniform random decision rule $\pi_u(\mathbf{X})$ by changing a mixture parameter $\alpha$, i.e., $\pi(\mathbf{X}) = \alpha\pi_d(\mathbf{X}) + (1 - \alpha)\pi_u(\mathbf{X})$. The candidates of the mixture parameter $\alpha$ are $\{0.6, 0.3, 0.0\}$.

Case 2: The potential outcome $Y(1)$ follows a Bernoulli distribution with probability of success $1/\{1 + \exp(X_1 + X_2 + X_3)\}$. The action $A$ is generated from $A \sim \text{Bernoulli}\{\varphi(\mathbf{X}, Y(1))\}$, and $\text{logit}\{\varphi(\mathbf{X}, Y(1))\} = 1/[1 - \exp\{0.5 + X_1 + X_2 + 0.5Y(1)\}]$. Thus, $X_3$ is the shadow variable. We construct three different evaluation decision rules as mixtures of a deterministic decision rule $\pi_d(\mathbf{X}) = \mathbb{I}(X_1 + X_2 + X_3 < 0)$ and the uniform random decision rule $\pi_u(\mathbf{X})$ by changing a mixture parameter $\alpha$, i.e., $\pi(\mathbf{X}) = \alpha\pi_d(\mathbf{X}) + (1 - \alpha)\pi_u(\mathbf{X})$. The candidates of the mixture parameter $\alpha$ are $\{0.7, 0.4, 0.0\}$.

For both cases, the true value function for each evaluation decision rule is obtained by generating a large sample $\{\mathbf{X}_i, Y_i(1)\}_{i=1}^{N}$ with size $N = 10^5$ and applying the empirical version of $V_1(\pi) = \mathbb{E}[Y(1)\pi(\mathbf{X})]$. We consider a correctly specified logistic regression model for $\varphi(\eta)$. We obtain $\widehat{\eta}_{\text{naive}}$ using $g(\mathbf{x}; \eta) = (1, x_1, x_2, x_3)^T$. We obtain the efficient estimators $\widehat{\eta}_{\text{eff}}$ and $\widehat{V}_{\text{eff}}(\pi)$ using the approach introduced in Section 5. Specifically, in case 1, all the regressions with pseudo-outcomes are using random forest (RF) models. In case 2, we estimate $\mathbb{P}(Y = 1 \mid \mathbf{X}, A = 1)$ using a generalized additive model (GAM). For the DR estimator, we estimate $w(\mathbf{x})$ using GAM in both cases. We estimate $\mathbb{E}(y \mid \mathbf{x})$ using RF in case 1 and using GAM in case 2.

We consider samples with size $n = 1000, 2000$. For each case, we conduct 500 replications. The root-mean-square error (RMSE), the standard deviation (SD), and the bias results for cases 1 and 2 are reported in Table 1 and Table 2.

Table 1: Decision evaluation results for case 1: (a) $0.0\pi_d + 1.0\pi_u$, (b) $0.3\pi_d + 0.7\pi_u$, (c) $0.6\pi_d + 0.4\pi_u$.

| | (a) | | | (b) | | | (c) | | |
|---|---|---|---|---|---|---|---|---|---|
| | RMSE | SD | Bias | RMSE | SD | Bias | RMSE | SD | Bias |
| | | | | | $n = 1000$ | | | | |
| $\widehat{V}_{\text{eff}}$ | 0.3512 | 0.3480 | 0.0468 | 0.5509 | 0.5483 | 0.0530 | 0.7999 | 0.7977 | 0.0591 |
| $\widehat{V}_{\text{IPW-naive}}$ | 0.7893 | 0.7890 | -0.0229 | 0.8279 | 0.8278 | -0.0127 | 0.8740 | 0.8740 | -0.0024 |
| $\widehat{V}_{\text{IPW-eff}}$ | 0.6172 | 0.6119 | 0.0807 | 0.8426 | 0.8387 | 0.0809 | 1.0852 | 1.0822 | 0.0810 |
| $\widehat{V}_{\text{DR}}$ | 0.4421 | 0.1559 | 0.4138 | 0.4371 | 0.1842 | 0.3964 | 0.4364 | 0.2162 | 0.3790 |
| | | | | | $n = 2000$ | | | | |
| $\widehat{V}_{\text{eff}}$ | 0.2003 | 0.1985 | 0.0274 | 0.2016 | 0.2005 | 0.0209 | 0.2169 | 0.2165 | 0.0143 |
| $\widehat{V}_{\text{IPW-naive}}$ | 0.7057 | 0.7026 | -0.0662 | 0.7363 | 0.7341 | -0.0575 | 0.7733 | 0.7718 | -0.0489 |
| $\widehat{V}_{\text{IPW-eff}}$ | 0.2563 | 0.2539 | 0.0353 | 0.2771 | 0.2761 | 0.0228 | 0.3121 | 0.3119 | 0.0103 |
| $\widehat{V}_{\text{DR}}$ | 0.3647 | 0.1077 | 0.3485 | 0.3538 | 0.1245 | 0.3312 | 0.3455 | 0.1444 | 0.3139 |

We have the following observations. $\widehat{V}_{\text{eff}}$, $\widehat{V}_{\text{IPW-naive}}$, and $\widehat{V}_{\text{IPW-eff}}$ are nearly unbiased with sample size $n = 1000, 2000$. However, $\widehat{V}_{\text{DR}}$ has a significantly larger bias when compared to other

Table 2: Decision evaluation results for case 2. (a) $0.0\pi_d + 1.0\pi_u$, (b) $0.4\pi_d + 0.6\pi_u$, (c) $0.7\pi_d + 0.3\pi_u$.

| | (a) | | | (b) | | | (c) | | |
| --- | --- | --- | --- | --- | --- | --- | --- | --- | --- |
| | RMSE | SD | Bias | RMSE | SD | Bias | RMSE | SD | Bias |
| | | | | $n = 1000$ | | | | | |
| $\widehat{V}_{\text{eff}}$ | 0.0172 | 0.0172 | -0.0005 | 0.0207 | 0.0207 | -0.0008 | 0.0239 | 0.0239 | -0.0011 |
| $\widehat{V}_{\text{IPW}-\text{naive}}$ | 0.0204 | 0.0204 | -0.0001 | 0.0246 | 0.0246 | -0.0003 | 0.0282 | 0.0282 | -0.0005 |
| $\widehat{V}_{\text{IPW}-\text{eff}}$ | 0.0179 | 0.0179 | -0.0006 | 0.0219 | 0.0219 | -0.0009 | 0.0254 | 0.0253 | -0.0012 |
| $\widehat{V}_{\text{DR}}$ | 0.0196 | 0.0097 | 0.0170 | 0.0223 | 0.0124 | 0.0185 | 0.0248 | 0.0152 | 0.0196 |
| | | | | $n = 2000$ | | | | | |
| $\widehat{V}_{\text{eff}}$ | 0.0119 | 0.0119 | -0.0005 | 0.0142 | 0.0142 | -0.0009 | 0.0163 | 0.0163 | -0.0013 |
| $\widehat{V}_{\text{IPW}-\text{naive}}$ | 0.0141 | 0.0141 | -0.0003 | 0.0167 | 0.0167 | -0.0006 | 0.0190 | 0.0190 | -0.0009 |
| $\widehat{V}_{\text{IPW}-\text{eff}}$ | 0.0122 | 0.0122 | -0.0004 | 0.0148 | 0.0147 | -0.0007 | 0.0171 | 0.0170 | -0.0009 |
| $\widehat{V}_{\text{DR}}$ | 0.0179 | 0.0069 | 0.0166 | 0.0198 | 0.0087 | 0.0178 | 0.0215 | 0.0106 | 0.0187 |

estimators. This is because the NUC assumption is violated in this setting. Among three consistent estimators $\widehat{V}_{\text{eff}}, \widehat{V}_{\text{IPW}-\text{naive}}$, and $\widehat{V}_{\text{IPW}-\text{eff}}$, $\widehat{V}_{\text{eff}}$ has the smallest standard deviation and RMSE, which is expected. One interesting observation is that for case 1, when sample size $n = 1000$, the standard deviations of $\widehat{V}_{\text{IPW}-\text{naive}}$ with decision rules (b) and (c) are smaller than those of $\widehat{V}_{\text{IPW}-\text{eff}}$. One possible reason is that when the sample size is small, the performance of nonparametric regressions with pseudo-outcomes may have larger variation. As the sample size increases, the standard deviations and RMSEs of three consistent estimators $\widehat{V}_{\text{eff}}, \widehat{V}_{\text{IPW}-\text{naive}}$, and $\widehat{V}_{\text{IPW}-\text{eff}}$ become smaller.

**Decision Learning:** We consider the same covariates as those used in decision evaluation. The potential outcome is generated by $Y(1) = 8X_1 - 6X_1^2 - 4X_2 + 2X_3^2 + \epsilon$, where $\epsilon$ is generated from a normal distribution with mean 0 and standard deviation 0.25. The action $A$ is generated from $A \sim \text{Bernoulli}\varphi(\mathbf{X}, Y(1)) = 1/[1 + \exp\{0.5 - X_1 - X_2 - 0.15Y(1)\}]$. We construct four estimators following the same procedure as in decision evaluation. We use a tree-based classification algorithm introduced in Zhou et al. (2023) and focus on depth-2 decision trees for illustration. To evaluate and compare the performance of estimated optimal decision rules obtained by different methods, we compute the corresponding value functions and percentages of making correct decisions (PCD). Again, we generate a large sample $\{\mathbf{X}_i, Y_i(1)\}_{i=1}^{N}$ with size $N = 10^5$. For a fixed decision rule $\pi$, its value function is computed using the empirical version of $V_1(\pi) = \mathbb{E}[Y(1)\pi(\mathbf{X})]$. We then maximize the value function and obtain the oracle optimal depth-2 decision tree, denoted as $\pi^*$. For each estimated optimal decision rule $\widehat{\pi}$, its associated value function is computed using the generated large sample and the PCD is computed by $N^{-1}\sum_{i=1}^{N} |\widehat{\pi}(\mathbf{X}_i) - \pi^*(\mathbf{X}_i)|$. We report the value and PCD results for the decision rules obtained by different methods in Figure 1. We observe that the decision rule obtained by our proposed method has best performance compared with other methods, in terms of values and PCDs. For our proposed method, as the sample size increases, the means of values become larger, PCDs get close to 1, and the standard deviations of values and PCDs become smaller.

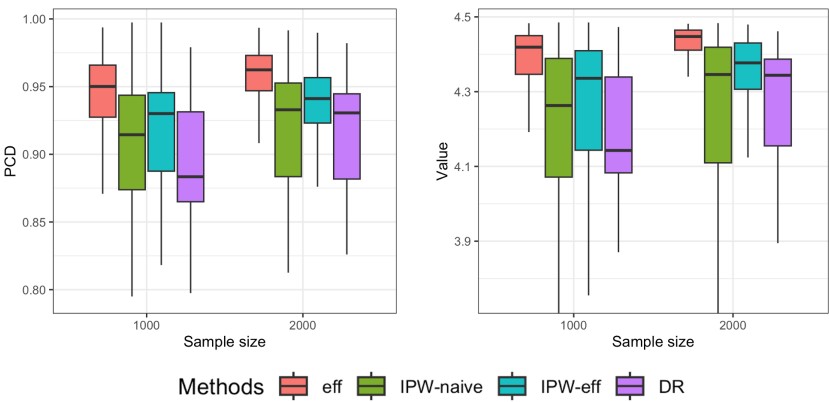

Figure 1: The values and PCDs of estimated optimal decision rules.

## 6.2 REAL DATA APPLICATION

In this section, we apply our method to a loan application dataset from a fintech company. A simulated dataset based on the real data is available upon request. The fintech lender aims to provide short-term credit to young salaried professionals by using their mobile and social footprints to determine their credit-worthiness. To get a loan, a customer needs to download the lending app, submit all the requisite details and documentation, and give permission to the lender to gather additional information from the smartphone, such as the number of apps and SMSs. We obtained data from the lending firm for all loans granted from February 2016 to November 2018. There are 42,777 customers in total. We select a set of covariates $\mathbf{X}$, which includes the applicants' age, salary, loan amount, CIBIL credit score, number of apps, number of SMSs, number of contacts, and number of social connections. The action $A$ are whether or not the lender approves the loan applications. The outcome $Y$ is defined as 1 if the loan is repaid, and -1 if the applicant defaults on the loan. We conduct hypothesis testing, and our analysis reveals no significant evidence suggesting that the number of social connections violates Assumption 4.1. Therefore, we consider it as a SV.

We randomly sample the training data with a size 3000 and 5000. We compare the four estimators introduced in Section 6.1, which are constructed using the same procedure for the binary outcome. Specifically, we estimate $\mathbb{E}(Y \mid \mathbf{X})$ for the DR method and $\mathbb{P}(Y \mid \mathbf{X}, A = 1)$ for the proposed method using GAM. For the DR method, we estimate $w(\mathbf{X})$ using GAM as well. We consider a logistic regression model for $\varphi(\eta)$ that uses all covariates (excluding the SV) and the potential outcome as predictors. We obtain $\widehat{\eta}_{\text{naive}}$ using $g(\mathbf{X}, \eta) = (1, \mathbf{X}^T)^T$. We use the same classification algorithm as in the synthetic scenarios to estimate the optimal decision rule. The proposed efficient estimator over the entire dataset is used as the testing value. The training-testing procedure is repeated 100 times. We report the results of testing values in Figure 2. We observe that the average value of proposed method is much larger than those of other three methods, while the variability of proposed method is smaller. This implies the proposed method has better performance than other three methods.

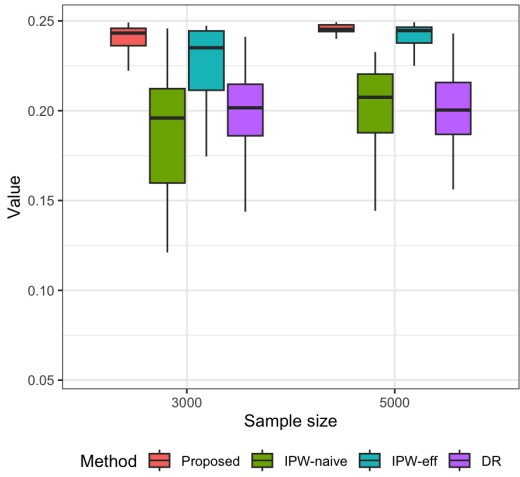

Figure 2: The boxplots of testing values under estimated optimal decision rules by different methods.

## 7 CONCLUSION

In this paper, we propose a novel framework for causal decision making under the one-sided feedback setting. Specifically, we define a new value function for this task and provide identification leveraging SVs, without assuming NUC. We develop efficient evaluation and learning methods motivated by the semiparametric theory. Numerical experiments and a real-world data application demonstrate the empirical performance of our proposed methods. Although this work focuses on the contextual bandits setting, our method has significant potential for extension to many semi-supervised learning tasks (Hu et al., 2022; Sportisse et al., 2023) and generative models (Ma & Zhang, 2021; Ipsen et al., 2021) with non-random missing data.

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

# A   Technical Proofs

## A.1   Proof of Theorem 4.3

*Proof.*

$$
\begin{aligned}
&\mathbb{E}\{Y(1) \mid \mathbf{X} = \mathbf{x}\} \\
=&\mathbb{E}\{Y(1) \mid \mathbf{X} = \mathbf{x}, A = 1\}w(\mathbf{x}) + \mathbb{E}\{Y(1) \mid \mathbf{X} = \mathbf{x}, A = 0\}\{1 - w(\mathbf{x})\} \\
=&w(\mathbf{x})\left\{\int yf(y \mid \mathbf{x}, 1)dy\right\} + \{1 - w(\mathbf{x})\}\left\{\int yf(y \mid \mathbf{x}, 0)dy\right\} \\
=&w(\mathbf{x})\left\{\int yf(y \mid \mathbf{x}, 1)dy\right\} + \left\{\int y\{1 - w(\mathbf{x})\}f(y \mid \mathbf{x}, 0)dy\right\} \\
=&w(\mathbf{x})\left\{\int yf(y \mid \mathbf{x}, 1)dy\right\} + \left\{\int yf(y \mid \mathbf{x}, 1)\left[\frac{f(y \mid \mathbf{x}, 0)\{1 - w(\mathbf{x})\}}{f(y \mid \mathbf{x}, 1)}\right]dy\right\} \\
=&w(\mathbf{x})\left\{\int yf(y \mid \mathbf{x}, 1)dy\right\} + \left\{\int yf(y \mid \mathbf{x}, 1)\left[w(\mathbf{x})\left\{\frac{1}{\varphi(\mathbf{x}, y)} - 1\right\}\right]dy\right\} \\
=&w(\mathbf{x})\left\{\int yf(y \mid \mathbf{x}, 1)dy\right\} + w(\mathbf{x})\left\{\int yf(y \mid \mathbf{x}, 1)\left[\left\{\frac{1}{\varphi(\mathbf{x}, y)} - 1\right\}\right]dy\right\} \\
=&w(\mathbf{x})\int y\frac{f(y \mid \mathbf{x}, 1)}{\varphi(\mathbf{x}, y)}dy.
\end{aligned}
$$

Therefore,

$$
\begin{aligned}
V_1(\pi) &= \mathbb{E}\{Y(1)\pi(\mathbf{X})\} \\
&=\mathbb{E}\left(\mathbb{E}[\{Y(1)\pi(\mathbf{X})\} \mid \mathbf{X}]\right) \\
&= \int f(\mathbf{x})\pi(\mathbf{x})\mathbb{E}\{Y(1) \mid \mathbf{X} = \mathbf{x}\}d\mathbf{x} \\
&= \int f(\mathbf{x})w(\mathbf{x})\left\{\int y\frac{f(y \mid \mathbf{x}, 1)}{\varphi(\mathbf{x}, y)}dy\right\}\pi(\mathbf{x})d\mathbf{x}.
\end{aligned}
$$

To identify $V_1(\pi)$, we need to identify $f(\mathbf{x})$, $w(\mathbf{x})$, $f(y \mid \mathbf{x}, 1)$, and $\varphi(\mathbf{x}, y)$. The likelihood function for a single observation is

$$
f(\mathbf{x})w(\mathbf{x})^a\{1 - w(\mathbf{x})\}^{1-a}f(y \mid \mathbf{x}, 1)^a.
$$

A key observation is that

$$
w(\mathbf{x})^{-1} = \int \frac{f(y \mid \mathbf{x}, 1)}{\varphi(\mathbf{x}, y)}dy.
$$

Under Assumption 4.1, $\varphi(\mathbf{x}, y) = \mathbb{P}\{A = 1 \mid \mathbf{X} = \mathbf{x}, Y(1) = y\} = \mathbb{P}\{A = 1 \mid \mathbf{U} = \mathbf{u}, Y(1) = y\} = \varphi(\mathbf{u}, y)$, and the likelihood function becomes

$$
f(\mathbf{x})\left\{\int \frac{f(y \mid \mathbf{x}, 1)}{\varphi(\mathbf{u}, y)}dy\right\}^{-a}\left[1 - \left\{\int \frac{f(y \mid \mathbf{x}, 1)}{\varphi(\mathbf{u}, y)}dy\right\}^{-1}\right]^{1-a}f(y \mid \mathbf{x}, 1)^a.
$$

Assume we have two different sets of models $f(\mathbf{x})$, $f(y \mid \mathbf{x}, 1)$, $\varphi(\mathbf{u}, y)$, and $\tilde{f}(\mathbf{x})$, $\tilde{f}(y \mid \mathbf{x}, 1)$, $\tilde{\varphi}(\mathbf{u}, y)$, such that

$$
\begin{aligned}
&f(\mathbf{x})\left\{\int \frac{f(y \mid \mathbf{x}, 1)}{\varphi(\mathbf{u}, y)}dy\right\}^{-a}\left[1 - \left\{\int \frac{f(y \mid \mathbf{x}, 1)}{\varphi(\mathbf{u}, y)}dy\right\}^{-1}\right]^{1-a}f(y \mid \mathbf{x}, 1)^a \\
=&\tilde{f}(\mathbf{x})\left\{\int \frac{\tilde{f}(y \mid \mathbf{x}, 1)}{\tilde{\varphi}(\mathbf{u}, y)}dy\right\}^{-a}\left[1 - \left\{\int \frac{\tilde{f}(y \mid \mathbf{x}, 1)}{\tilde{\varphi}(\mathbf{u}, y)}dy\right\}^{-1}\right]^{1-a}\tilde{f}(y \mid \mathbf{x}, 1)^a. \quad (10)
\end{aligned}
$$

Taking $a = 0$ in (10), we have

$$f(\mathbf{x}) \left[ 1 - \left\{ \int \frac{f(y \mid \mathbf{x}, 1)}{\varphi(\mathbf{u}, y)} dy \right\}^{-1} \right] = \tilde{f}(\mathbf{x}) \left[ 1 - \left\{ \int \frac{\tilde{f}(y \mid \mathbf{x}, 1)}{\tilde{\varphi}(\mathbf{u}, y)} dy \right\}^{-1} \right]. \tag{11}$$

Taking $a = 1$ and taking integration with respect to $Y(1)$ on both sides of the above equation, we have

$$f(\mathbf{x}) \left\{ \int \frac{f(y \mid \mathbf{x}, 1)}{\varphi(\mathbf{u}, y)} dy \right\}^{-1} = \tilde{f}(\mathbf{x}) \left\{ \int \frac{\tilde{f}(y \mid \mathbf{x}, 1)}{\tilde{\varphi}(\mathbf{u}, y)} dy \right\}^{-1}. \tag{12}$$

By Equations (11) and (12), we have

$$f(\mathbf{x}) = \tilde{f}(\mathbf{x}) \quad \text{and} \quad \int \frac{f(y \mid \mathbf{x}, 1)}{\varphi(\mathbf{u}, y)} dy = \int \frac{\tilde{f}(y \mid \mathbf{x}, 1)}{\tilde{\varphi}(\mathbf{u}, y)} dy.$$

Taking $a = 1$ in (10), we have

$$f(\mathbf{x}) \left\{ \int \frac{f(y \mid \mathbf{x}, 1)}{\varphi(\mathbf{u}, y)} dy \right\}^{-1} f(y \mid \mathbf{x}, 1) = \tilde{f}(\mathbf{x}) \left\{ \int \frac{\tilde{f}(y \mid \mathbf{x}, 1)}{\tilde{\varphi}(\mathbf{u}, y)} dy \right\}^{-1} \tilde{f}(y \mid \mathbf{x}, 1).$$

Thus, we have

$$f(y \mid \mathbf{x}, 1) = \tilde{f}(y \mid \mathbf{x}, 1).$$

Finally, from

$$\int \frac{f(y \mid \mathbf{x}, 1)}{\varphi(\mathbf{u}, y)} dy = \int \frac{f(y \mid \mathbf{x}, 1)}{\tilde{\varphi}(\mathbf{u}, y)} dy,$$

and Assumption 4.2, we have

$$\varphi(\mathbf{u}, y) = \tilde{\varphi}(\mathbf{u}, y).$$

Thus, $f(\mathbf{x})$, $w(\mathbf{x})$, $f(y \mid \mathbf{x}, 1)$, and $\varphi(\mathbf{x}, y)$ are all identified. The value function $V_1(\pi)$ is then identified. $\qquad\square$

## A.2 PROOF OF THEOREM 4.4

*Proof.* Let $\mathbf{O} = \{AY, A, \mathbf{X}\}$ summarize the vector of observed variables with the likelihood factorized as

$$f(\mathbf{O}) = f(\mathbf{X}) w(\mathbf{X})^A \{1 - w(\mathbf{X})\}^{1-A} f(Y \mid \mathbf{X}, A = 1)^A.$$

We consider a one-dimensional parametric submodel $f_{\theta_1}(\mathbf{X})$ for $f(\mathbf{X})$, and a one-dimensional parametric submodel $f_{\theta_2}(Y \mid \mathbf{X}, A = 1)$ for $f(Y \mid \mathbf{X}, A = 1)$, respectively. The submodel $f_{\theta_1}(\mathbf{X})$ contains the true model $f(\mathbf{X})$ at $\theta_1 = 0$, i.e., $f_{\theta_1}(\mathbf{X}) \mid_{\theta_1=0} = f(\mathbf{X})$. Similarly, the submodel $f_{\theta_2}(Y \mid \mathbf{X}, A = 1)$ contains the true model $f(Y \mid \mathbf{X}, A = 1)$ at $\theta_2 = 0$, i.e., $f_{\theta_2}(Y \mid \mathbf{X}, A = 1) \mid_{\theta_2=0} = f(Y \mid \mathbf{X}, A = 1)$. The submodel for the likelihood is

$$f_{\theta_1, \theta_2}(\mathbf{O}) = f_{\theta_1}(\mathbf{X}) w_{\theta_2}(\mathbf{X})^A \{1 - w_{\theta_2}(\mathbf{X})\}^{1-A} f_{\theta_2}(Y \mid \mathbf{X}, A = 1)^A.$$

$$\frac{\partial \log f_{\theta_1, \theta_2}(\mathbf{O})}{\partial \theta_1} = \frac{\partial \log f_{\theta_1}(\mathbf{X})}{\partial \theta_1},$$

$$\frac{\partial \log f_{\theta_1, \theta_2}(\mathbf{O})}{\partial \theta_2} = A \frac{\partial \log f_{\theta_2}(Y \mid \mathbf{X}, A = 1)}{\partial \theta_2} + \frac{w_{\theta_2}(\mathbf{X}) - A}{1 - w_{\theta_2}(\mathbf{X})} \mathbb{E} \left\{ \frac{\partial \log f_{\theta_2}(Y \mid \mathbf{X}, A = 1)}{\partial \theta_2} \mid \mathbf{X} \right\}.$$

By the semiparametric theory (Bickel et al., 1993; Tsiatis, 2006), we have the nuisance tangent spaces

$$\Lambda_1 = \left[ h_1(\mathbf{X}) : \mathbb{E}\{h_1(\mathbf{X}) = 0\} \right],$$

$$\Lambda_2 = \left[ A h_2(\mathbf{X}, Y(1)) + \frac{w(\mathbf{X}) - A}{1 - w(\mathbf{X})} \mathbb{E}\{h_2(\mathbf{X}, Y(1)) \mid \mathbf{X}\} : \mathbb{E}\{h_2(\mathbf{X}, Y(1)) \mid \mathbf{X}, A = 1\} = 0 \right].$$

It is easy to verify that $\Lambda_1 \perp \Lambda_2$. Consider a generic mean zero element in $\Lambda_\perp$, $Ag_1(\mathbf{X}, Y(1)) + (1 - A)g_2(\mathbf{X})$. Since $\Lambda_1 \perp \Lambda_\perp$, for any measurable mean zero function $h_1(\mathbf{X})$, we have

$$\mathbb{E}[\{Ag_1(\mathbf{X}, Y(1)) + (1 - A)g_2(\mathbf{X})\}h_1(\mathbf{X})]$$
$$=\mathbb{E}(\mathbb{E}[\{Ag_1(\mathbf{X}, Y(1)) + (1 - A)g_2(\mathbf{X})\}h_1(\mathbf{X}) \mid \mathbf{X}])$$
$$=\mathbb{E}([w(\mathbf{X})\mathbb{E}\{g_1(\mathbf{X}, Y(1)) \mid \mathbf{X}, A = 1\} + \{1 - w(\mathbf{X})\}g_2(\mathbf{X})]h_1(\mathbf{X}))$$
$$=0.$$

Therefore, $w(\mathbf{X})\mathbb{E}\{g_1(\mathbf{X}, Y(1)) \mid \mathbf{X}, A = 1\} + \{1 - w(\mathbf{X})\}g_2(\mathbf{X})$ is a constant and we denote it as $c$. Since $Ag_1(\mathbf{X}, Y(1)) + (1 - A)g_2(\mathbf{X})$ is mean zero, we have

$$\mathbb{E}\{Ag_1(\mathbf{X}, Y(1)) + (1 - A)g_2(\mathbf{X})\}$$
$$=\mathbb{E}[w(\mathbf{X})\mathbb{E}\{g_1(\mathbf{X}, Y(1)) \mid \mathbf{X}, A = 1\} + \{1 - w(\mathbf{X})\}g_2(\mathbf{X})]$$
$$=\mathbb{E}(c) = 0.$$

Therefore, we have

$$w(\mathbf{X})\mathbb{E}\{g_1(\mathbf{X}, Y(1)) \mid \mathbf{X}, A = 1\} + \{1 - w(\mathbf{X})\}g_2(\mathbf{X}) = 0. \tag{13}$$

Since $\Lambda_2 \perp \Lambda_\perp$, we have

$$\mathbb{E}\left(\{Ag_1(\mathbf{X}, Y(1)) + (1 - A)g_2(\mathbf{X})\}\left[Ah_2(\mathbf{X}, Y(1)) + \frac{w(\mathbf{X}) - A}{1 - w(\mathbf{X})}\mathbb{E}\{h_2(\mathbf{X}, Y(1)) \mid \mathbf{X}\}\right]\right)$$
$$=\mathbb{E}\left[w(\mathbf{X})\mathbb{E}\{g_1(\mathbf{X}, Y(1))h_2(\mathbf{X}, Y(1)) \mid \mathbf{X}, A = 1\} + g_2(\mathbf{X})\mathbb{E}\{h_2(\mathbf{X}, Y(1)) \mid \mathbf{X}\}\right]$$
$$=\mathbb{E}\left[w(\mathbf{X})\mathbb{E}\{g_1(\mathbf{X}, Y(1))h_2(\mathbf{X}, Y(1)) \mid \mathbf{X}, A = 1\} + w(\mathbf{X})g_2(\mathbf{X})\mathbb{E}\left\{\frac{h_2(\mathbf{X}, Y(1))}{\varphi(\eta)} \mid \mathbf{X}, A = 1\right\}\right]$$
$$=\mathbb{E}\left(\mathbb{E}\left[w(\mathbf{X})\left\{g_1(\mathbf{X}, Y(1)) + \frac{g_2(\mathbf{X})}{\varphi(\eta)}\right\}h_2(\mathbf{X}, Y(1)) \mid \mathbf{X}, A = 1\right]\right)$$
$$=0.$$

Therefore, $g_1(\mathbf{X}, Y(1)) + \frac{g_2(\mathbf{X})}{\varphi(\eta)}$ is a function of $\mathbf{X}$ and we denote it as $k(\mathbf{X})$:

$$k(\mathbf{X}) = g_1(\mathbf{X}, Y(1)) + \frac{g_2(\mathbf{X})}{\varphi(\eta)}.$$

Taking the conditional expectation on both sides, and by (13), we have

$$k(\mathbf{X}) = \mathbb{E}\{g_1(\mathbf{X}, Y(1)) \mid \mathbf{X}, A = 1\} + \frac{g_2(\mathbf{X})}{w(\mathbf{X})} = g_2(\mathbf{X}).$$

Therefore, we have

$$g_2(\mathbf{X}) = g_1(\mathbf{X}, Y(1)) + \frac{g_2(\mathbf{X})}{\varphi(\eta)}.$$

Thus,

$$Ag_1(\mathbf{X}, Y(1)) + (1 - A)g_2(\mathbf{X}) = \frac{\varphi(\eta) - A}{\varphi(\eta)}g_1(\mathbf{X}),$$

and $\Lambda_\perp = \left\{\frac{\varphi(\eta) - A}{\varphi(\eta)}g_1(\mathbf{X})\right\}$. This completes the proof. $\square$

## A.3 PROOF OF THEOREM 4.5

*Proof.* The score function for $\eta$ is

$$S_\eta = \frac{A - w(\mathbf{X})}{1 - w(\mathbf{X})}\mathbb{E}\left\{\frac{\dot{\varphi}(\eta)}{\varphi(\eta)} \mid \mathbf{X}\right\}.$$

The efficient score for $\eta$ is the projection of the score function $S_\eta$ onto the space $\Lambda_\perp$. Notice that $S_\eta \perp \Lambda_1$. Therefore, we can write

$$\frac{A - w(\mathbf{X})}{1 - w(\mathbf{X})}\mathbb{E}\left\{\frac{\dot{\varphi}(\eta)}{\varphi(\eta)} \mid \mathbf{X}\right\} = \underbrace{Ab(\mathbf{X}, Y(1)) + \frac{w(\mathbf{X}) - A}{1 - w(\mathbf{X})}\mathbb{E}\{b(\mathbf{X}, Y(1)) \mid \mathbf{X}\}}_{\in \Lambda_2} + \underbrace{\frac{\varphi(\eta) - A}{\varphi(\eta)}c(\mathbf{X})}_{\Lambda_\perp},$$

$$\tag{14}$$

where $\mathbb{E}\{b(\mathbf{X}, Y(1)) \mid \mathbf{X}, A = 1\} = 0$. Let $A = 1$ in (14), we have

$$\mathbb{E}\left\{\frac{\dot{\varphi}(\eta)}{\varphi(\eta)} \mid \mathbf{X}\right\} = b(\mathbf{X}, Y(1)) - \mathbb{E}\{b(\mathbf{X}, Y(1)) \mid \mathbf{X}\} + \frac{\varphi(\eta) - 1}{\varphi(\eta)} c(\mathbf{X}).$$

By taking $\mathbb{E}(\cdot \mid \mathbf{X})$ on both sides, we have

$$c(\mathbf{X}) = \frac{\mathbb{E}\left\{\frac{\dot{\varphi}(\eta)}{\varphi(\eta)} \mid \mathbf{X}\right\}}{1 - \mathbb{E}\left\{\frac{1}{\varphi(\eta)} \mid \mathbf{X}\right\}} = \frac{\mathbb{E}\left\{\frac{\dot{\varphi}(\eta)}{\varphi(\eta)^2} \mid \mathbf{X}, A = 1\right\}}{\mathbb{E}\left\{\frac{\varphi(\eta) - 1}{\varphi(\eta)^2} \mid \mathbf{X}, A = 1\right\}}.$$

Therefore,

$$S_{\eta, \text{eff}} = \frac{\varphi(\eta) - A}{\varphi(\eta)} \frac{\mathbb{E}\left\{\frac{\dot{\varphi}(\eta)}{\varphi(\eta)^2} \mid \mathbf{X}, A = 1\right\}}{\mathbb{E}\left\{\frac{\varphi(\eta) - 1}{\varphi(\eta)^2} \mid \mathbf{X}, A = 1\right\}}.$$

Let $A = 0$ in (14), we can further derive that

$$b(\mathbf{X}, Y(1)) = \left\{\frac{1}{\varphi(\eta)} - \frac{1}{w(\mathbf{X})}\right\} c(\mathbf{X}).$$

$\square$

## A.4 PROOF OF THEOREM 4.6

*Proof.* We consider a one-dimensional parametric submodel $f_\alpha(\mathbf{X})$ for $f(\mathbf{X})$, and a one-dimensional parametric submodel $f_\beta(Y \mid \mathbf{X}, A = 1)$ for $f(Y \mid \mathbf{X}, A = 1)$, respectively. The submodel $f_\alpha(\mathbf{X})$ contains the true model $f(\mathbf{X})$ at $\alpha = \alpha_0$, i.e., $f_{\alpha_0}(\mathbf{X}) = f(\mathbf{X})$. Similarly, the submodel $f_\beta(Y \mid \mathbf{X}, A = 1)$ contains the true model $f(Y \mid \mathbf{X}, A = 1)$ at $\beta = \beta_0$, i.e., $f_{\beta_0}(Y \mid \mathbf{X}, A = 1) = f(Y \mid \mathbf{X}, A = 1)$. Let $\theta = (\alpha, \beta)$. The submodel for the likelihood can be represented as

$$f_{\theta, \eta}(\mathbf{O}) = f_\alpha(\mathbf{X})\{w_{\beta, \eta}(\mathbf{X})\}^A f_\beta(Y \mid \mathbf{X}, A = 1)\{1 - w_{\beta, \eta}(\mathbf{X})\}^{1-A},$$

which contains the true model at $\theta_0 = (\alpha_0, \beta_0)$. For the ease of exposition, we write $V_1(\pi)$ as $V(\pi)$. We use $\theta$ in the subscript to denote the quantity with respect to the submodel, e.g., $V_\theta(\pi)$ is the value of $V(\pi)$ in the submodel.

Let

$$S_{\alpha_0} = \left.\frac{\partial \log f_\theta(\mathbf{O})}{\partial \alpha}\right|_{\theta = \theta_0} = \left.\frac{\partial \log f_\alpha(\mathbf{X})}{\partial \alpha}\right|_{\alpha = \alpha_0},$$

$$S_{\beta_0} = \left.\frac{\partial \log f_\theta(\mathbf{O})}{\partial \beta}\right|_{\theta = \theta_0} = A\left.\frac{\partial \log f_\beta(Y \mid \mathbf{X}, A = 1)}{\partial \beta}\right|_{\beta = \beta_0} + \frac{w(\mathbf{X}) - A}{1 - w(\mathbf{X})}\mathbb{E}\left\{\left.\frac{\partial \log f_\beta(Y \mid \mathbf{X}, A = 1)}{\partial \beta}\right|_{\beta = \beta_0} \mid \mathbf{X}\right\},$$

$$S_\eta = \left.\frac{\partial \log f_\theta(\mathbf{O})}{\partial \eta}\right|_{\theta = \theta_0} = \frac{A - w(\mathbf{X})}{1 - w(\mathbf{X})}\mathbb{E}\left\{\frac{\partial \log \varphi(\eta)}{\partial \eta} \mid \mathbf{X}\right\}.$$

Let $s_{\beta_0} = \left.\frac{\partial \log f_\beta(Y \mid \mathbf{X}, A = 1)}{\partial \beta}\right|_{\beta = \beta_0}$ and $s_\eta = \frac{\partial \log \varphi(\eta)}{\partial \eta}$.

By the semiparametric theory, the EIF for $V(\pi)$ must have the form

$$\phi_{\text{eff}} = \underbrace{h_1^*(\mathbf{X})}_{\in \Lambda_1} + \underbrace{Ah_2^*(\mathbf{X}) + \frac{w(\mathbf{X}) - A}{1 - w(\mathbf{X})}\mathbb{E}\{h_2^*(\mathbf{X}, Y(1)) \mid \mathbf{X}\}}_{\in \Lambda_2} + \underbrace{\boldsymbol{D}^T S_{\eta, \text{eff}}}_{\in \Lambda_\perp},$$

where $\mathbb{E}\{h_1^*(\mathbf{X}) = 0\}, \mathbb{E}\{h_2^*(\mathbf{X}, Y(1)) \mid \mathbf{X}, A = 1\} = 0$, and $\boldsymbol{D}$ is a vector with the same dimension as $\eta$. The EIF $\phi_{\text{eff}}$ for $V(\pi)$ must satisfy

$$\partial V_\theta(\pi)/\partial \alpha|_{\theta = \theta_0} = \mathbb{E}(\phi_{\text{eff}} S_{\alpha_0}),$$
$$\partial V_\theta(\pi)/\partial \beta|_{\theta = \theta_0} = \mathbb{E}(\phi_{\text{eff}} S_{\beta_0}),$$
$$\partial V_\theta(\pi)/\partial \eta^T|_{\theta = \theta_0} = \mathbb{E}(\phi_{\text{eff}} S_\eta^T).$$

(I)

$$\partial V_\theta(\pi)/\partial\alpha \mid_{\theta=\theta_0} = \mathbb{E}\left[\pi(\mathbf{X})w(\mathbf{X})\mathbb{E}\left\{\frac{Y}{\varphi(\eta)}\mid\mathbf{X},A=1\right\}S_{\alpha_0}\right],$$

$$\mathbb{E}(\phi_{\text{eff}}S_{\alpha_0}) = \mathbb{E}\{h_1^*(\mathbf{X})S_{\alpha_0}\}.$$

We have

$$h_1^*(\mathbf{X}) = \pi(\mathbf{X})w(\mathbf{X})\mathbb{E}\left\{\frac{Y}{\varphi(\eta)}\mid\mathbf{X},A=1\right\} - V(\pi).$$

(II)

$$\partial V_\theta(\pi)/\partial\beta \mid_{\theta=\theta_0} = \mathbb{E}\left[\pi(\mathbf{X})\{Y(1)-\mathbb{E}(Y(1)\mid\mathbf{X})\}s_{\beta_0}\right],$$

$$\mathbb{E}(\phi_{\text{eff}}S_{\beta_0}) = \mathbb{E}\left(\left[\varphi(\eta)h_2^*(\mathbf{X},Y(1)) + \frac{w(\mathbf{X})}{1-w(\mathbf{X})}\mathbb{E}\{h_2^*(\mathbf{X},Y(1))\mid\mathbf{X}\}\right]s_{\beta_0}\right).$$

$$\partial V_\theta(\pi)/\partial\beta \mid_{\theta=\theta_0} - \mathbb{E}(\phi_{\text{eff}}S_{\beta_0})$$
$$=\mathbb{E}\left(\left[\varphi(\eta)h_2^*(\mathbf{X},Y(1)) + \frac{w(\mathbf{X})}{1-w(\mathbf{X})}\mathbb{E}\{h_2^*(\mathbf{X},Y(1))\mid\mathbf{X}\} - \pi(\mathbf{X})\{Y(1)-\mathbb{E}\{Y(1)\mid\mathbf{X}\}\right]s_{\beta_0}\right)$$
$$=\mathbb{E}\left\{\mathbb{E}\left(\left[h_2^*(\mathbf{X},Y(1)) + \frac{w(\mathbf{X})}{1-w(\mathbf{X})}\frac{\mathbb{E}\{h_2^*(\mathbf{X},Y(1))\}\mid\mathbf{X}\}}{\varphi(\eta)} - \pi(\mathbf{X})\frac{Y(1)-\mathbb{E}\{Y(1)\mid\mathbf{X}\}}{\varphi(\eta)}\right]\varphi(\eta)s_{\beta_0}\right)\mid\mathbf{X}\right\}.$$

Since $\mathbb{E}\{\varphi(\eta)s_{\beta_0}\mid\mathbf{X}\}=0$, $h_2^*(\mathbf{X},Y(1)) + \frac{w(\mathbf{X})}{1-w(\mathbf{X})}\frac{\mathbb{E}\{h_2^*(\mathbf{X},Y(1))\}\mid\mathbf{X}\}}{\varphi(\eta)} - \pi(\mathbf{X})\frac{Y(1)-\mathbb{E}\{Y(1)\mid\mathbf{X}\}}{\varphi(\eta)}$ must be a function of $\mathbf{X}$ and we denote it as $m(\mathbf{X})$:

$$m(\mathbf{X}) = h_2^*(\mathbf{X},Y(1)) + \frac{w(\mathbf{X})}{1-w(\mathbf{X})}\frac{\mathbb{E}\{h_2^*(\mathbf{X},Y(1))\}\mid\mathbf{X}\}}{\varphi(\eta)} - \pi(\mathbf{X})\frac{Y(1)-\mathbb{E}\{Y(1)\mid\mathbf{X}\}}{\varphi(\eta)}. \quad (15)$$

Taking the conditional expectation on both sides, we have

$$m(\mathbf{X}) = \frac{\mathbb{E}\{h_2^*(\mathbf{X},Y(1))\mid\mathbf{X}\}}{1-w(\mathbf{X})}.$$

Therefore, we have

$$\frac{\mathbb{E}\{h_2^*(\mathbf{X},Y(1))\mid\mathbf{X}\}}{1-w(\mathbf{X})} = h_2^*(\mathbf{X},Y(1)) + \frac{w(\mathbf{X})}{1-w(\mathbf{X})}\frac{\mathbb{E}\{h_2^*(\mathbf{X},Y(1))\}\mid\mathbf{X}\}}{\varphi(\eta)} - \pi(\mathbf{X})\frac{Y(1)-\mathbb{E}\{Y(1)\mid\mathbf{X}\}}{\varphi(\eta)}.$$

Taking $\mathbb{E}(\cdot\mid\mathbf{X})$ on both sides,

$$\frac{\mathbb{E}\{h_2^*(\mathbf{X},Y(1))\mid\mathbf{X}\}}{1-w(\mathbf{X})}$$
$$=\mathbb{E}\{h_2^*(\mathbf{X},Y(1))\mid\mathbf{X}\} + \frac{w(\mathbf{X})}{1-w(\mathbf{X})}\mathbb{E}\{h_2^*(\mathbf{X},Y(1))\mid\mathbf{X}\}\mathbb{E}\{1/\varphi(\eta)\mid\mathbf{X}\}$$
$$- \pi(\mathbf{X})\left[\mathbb{E}\{Y(1)/\varphi(\eta)\mid\mathbf{X}\} - \mathbb{E}\{Y(1)\mid\mathbf{X}\}\mathbb{E}\{1/\varphi(\eta)\mid\mathbf{X}\}\right].$$

We have

$$\mathbb{E}\{h_2^*(\mathbf{X},Y(1))\mid\mathbf{X}\} = \pi(\mathbf{X})\frac{1-w(\mathbf{X})}{w(\mathbf{X})}\frac{\mathbb{E}\{Y(1)/\varphi(\eta)\mid\mathbf{X}\} - \mathbb{E}\{Y(1)\mid\mathbf{X}\}\mathbb{E}\{1/\varphi(\eta)\mid\mathbf{X}\}}{\mathbb{E}\{1/\varphi(\eta)\mid\mathbf{X}\}-1}. \quad (16)$$

By Equations (15) and (16),

$$h_2^*(\mathbf{X},Y(1)) = \pi(\mathbf{X})\left[\left\{\frac{1}{w(\mathbf{X})} - \frac{1}{\varphi(\eta)}\right\}\frac{\mathbb{E}\left\{\frac{Y(1)}{\varphi(\eta)}\mid\mathbf{X}\right\} - \mathbb{E}\{Y(1)\mid\mathbf{X}\}\mathbb{E}\left\{\frac{1}{\varphi(\eta)}\mid\mathbf{X}\right\}}{\mathbb{E}\{1/\varphi(\eta)\mid\mathbf{X}\}-1} + \frac{Y(1)-\mathbb{E}\{Y(1)\mid\mathbf{X}\}}{\varphi(\eta)}\right].$$

(III)

$$\partial V_\theta(\pi)/\partial \eta|_{\theta=\theta_0} = \mathbb{E}\left[\pi(\mathbf{X})\frac{\mathbb{E}\left\{Y(1)\frac{1-\varphi(\eta)}{\varphi(\eta)} \mid \mathbf{X}\right\}}{\mathbb{E}\left\{\frac{1-\varphi(\eta)}{\varphi(\eta)} \mid \mathbf{X}\right\}}\frac{\dot\varphi(\eta)}{\varphi(\eta)}\right] - \mathbb{E}\left\{\pi(\mathbf{X})Y(1)\frac{\dot\varphi(\eta)}{\varphi(\eta)}\right\}.$$

$$\mathbb{E}(\phi_{\text{eff}}S_\eta^T) = \boldsymbol{D}^T\mathbb{E}\{S_{\text{eff}}(\eta)S_{\text{eff}}(\eta)^T\}.$$

By $\partial V_\theta(\pi)/\partial \eta^T|_{\theta=\theta_0} = \mathbb{E}(\phi_{\text{eff}}S_\eta^T)$,

$$\boldsymbol{D} = \{\text{Var}(S_{\eta,\text{eff}})\}^{-1}\left(\mathbb{E}\left[\pi(\mathbf{X})\frac{\mathbb{E}\left\{\frac{1-\varphi(\eta)}{\varphi(\eta)^2}Y \mid \mathbf{X}, A=1\right\}}{\mathbb{E}\left\{\frac{1-\varphi(\eta)}{\varphi(\eta)^2} \mid \mathbf{X}, A=1\right\}}\frac{\dot\varphi(\eta)}{\varphi(\eta)}\right] - \mathbb{E}\left[\pi(\mathbf{X})\mathbb{E}\left\{\frac{\dot\varphi(\eta)}{\varphi(\eta)^2}Y \mid \mathbf{X}, A=1\right\}\right]\right).$$

By (I),(II), and (III), we complete the proof. $\qquad\square$

### A.5   PROOF OF THEOREM 5.2

*Proof.*

$$\mathbb{E}\left(\pi(\mathbf{X})\left[\frac{A}{\varphi(\eta)}Y + \left\{1 - \frac{A}{\varphi(\eta)}\right\}\frac{\mathbb{E}\left\{\frac{1-\varphi(\eta)}{\varphi(\eta)^2}Y \mid \mathbf{X}, 1\right\}}{\mathbb{E}\left\{\frac{1-\varphi(\eta)}{\varphi(\eta)^2} \mid \mathbf{X}, 1\right\}}\right]\right)$$

$$=\mathbb{E}\left\{\pi(\mathbf{X})\frac{A}{\varphi(\eta)}Y\right\}$$

$$=\mathbb{E}\left\{\pi(\mathbf{X})\frac{A}{\varphi(\eta)}AY(1)\right\}$$

$$=\mathbb{E}\left[\mathbb{E}\left\{\pi(\mathbf{X})\frac{A}{\varphi(\eta)}Y(1) \mid \mathbf{X}, Y(1)\right\}\right]$$

$$=\mathbb{E}\left[\pi(\mathbf{X})\frac{Y(1)}{\varphi(\eta)}\mathbb{E}\left\{A \mid \mathbf{X}, Y(1)\right\}\right]$$

$$=\mathbb{E}\left\{\pi(\mathbf{X})Y(1)\right\} = V_1(\pi).$$

Since a solution to Equation (7) is a root-$n$ estimator of $\eta$, by the strong law of large numbers and uniform consistency, we have $\widehat{V}_{\text{eff}}(\pi) = V_1(\pi) + o_p(1)$.

By Assumption 5.1 and the empirical process theory, we have

$$\mathbb{P}_n\left[\frac{\varphi(\widehat\eta_{\text{eff}}) - a}{\varphi(\widehat\eta_{\text{eff}})}\frac{\widehat{\mathbb{E}}\left\{\frac{\dot\varphi(\eta)}{\varphi(\eta)^2} \mid \mathbf{x}, 1\right\}}{\widehat{\mathbb{E}}\left\{\frac{\varphi(\eta)-1}{\varphi(\eta)^2} \mid \mathbf{x}, 1\right\}}\right] - \mathbb{P}_n\left[\frac{\varphi(\widehat\eta_{\text{eff}}) - a}{\varphi(\widehat\eta_{\text{eff}})}\frac{\mathbb{E}\left\{\frac{\dot\varphi(\eta)}{\varphi(\eta)^2} \mid \mathbf{x}, 1\right\}}{\mathbb{E}\left\{\frac{\varphi(\eta)-1}{\varphi(\eta)^2} \mid \mathbf{x}, 1\right\}}\right]$$

$$=\mathbb{P}\left[\frac{\varphi(\widehat\eta_{\text{eff}}) - a}{\varphi(\widehat\eta_{\text{eff}})}\frac{\widehat{\mathbb{E}}\left\{\frac{\dot\varphi(\eta)}{\varphi(\eta)^2} \mid \mathbf{x}, 1\right\}}{\widehat{\mathbb{E}}\left\{\frac{\varphi(\eta)-1}{\varphi(\eta)^2} \mid \mathbf{x}, 1\right\}}\right] - \mathbb{P}\left[\frac{\varphi(\widehat\eta_{\text{eff}}) - a}{\varphi(\widehat\eta_{\text{eff}})}\frac{\mathbb{E}\left\{\frac{\dot\varphi(\eta)}{\varphi(\eta)^2} \mid \mathbf{x}, 1\right\}}{\mathbb{E}\left\{\frac{\varphi(\eta)-1}{\varphi(\eta)^2} \mid \mathbf{x}, 1\right\}}\right] + o_p(n^{-1/2}). \quad (17)$$

For the ease of exposition, let $\mathbb{E}_1 = \mathbb{E}\left\{\frac{\dot{\varphi}(\eta)}{\varphi(\eta)^2} \mid \mathbf{x}, 1\right\}$ and $\mathbb{E}_2 = \mathbb{E}\left\{\frac{\varphi(\eta)-1}{\varphi(\eta)^2} \mid \mathbf{x}, 1\right\}$. By Assumptions 5.1, we have

$$
\begin{aligned}
&\left| \mathbb{P}\left\{ \frac{\varphi(\widehat{\eta}_{\text{eff}}) - a}{\varphi(\widehat{\eta}_{\text{eff}})} \frac{\widehat{\mathbb{E}}_1}{\widehat{\mathbb{E}}_2} \right\} - \mathbb{P}\left\{ \frac{\varphi(\widehat{\eta}_{\text{eff}}) - a}{\varphi(\widehat{\eta}_{\text{eff}})} \frac{\mathbb{E}_1}{\mathbb{E}_2} \right\} \right| \\
&= \left| \mathbb{P}\left[ \frac{\varphi(\widehat{\eta}_{\text{eff}}) - a}{\varphi(\widehat{\eta}_{\text{eff}})} \left\{ \frac{\widehat{\mathbb{E}}_1}{\widehat{\mathbb{E}}_2} - \frac{\mathbb{E}_1}{\mathbb{E}_2} \right\} \right] \right| \\
&= \left| \mathbb{P}\left[ \frac{\varphi(\widehat{\eta}_{\text{eff}}) - a}{\varphi(\widehat{\eta}_{\text{eff}})} \left\{ \frac{\widehat{\mathbb{E}}_1}{\widehat{\mathbb{E}}_2} - \frac{\mathbb{E}_1}{\widehat{\mathbb{E}}_2} + \frac{\mathbb{E}_1}{\widehat{\mathbb{E}}_2} - \frac{\mathbb{E}_1}{\mathbb{E}_2} \right\} \right] \right| \\
&= \left| \mathbb{P}\left[ \frac{\varphi(\widehat{\eta}_{\text{eff}}) - a}{\varphi(\widehat{\eta}_{\text{eff}})} \left\{ \frac{\widehat{\mathbb{E}}_1 - \mathbb{E}_1}{\widehat{\mathbb{E}}_2} + \frac{\mathbb{E}_1(\mathbb{E}_2 - \widehat{\mathbb{E}}_2)}{\mathbb{E}_2 \widehat{\mathbb{E}}_2} \right\} \right] \right| \\
&\leq O_p(n^{-1/2}) \times o_p(1) \\
&= o_p(n^{-1/2}).
\end{aligned}
\tag{18}
$$

By Equations (17) and (18), we have

$$
\mathbb{P}_n\left[ \frac{\varphi(\widehat{\eta}_{\text{eff}}) - a}{\varphi(\widehat{\eta}_{\text{eff}})} \frac{\widehat{\mathbb{E}}\left\{\frac{\dot{\varphi}(\eta)}{\varphi(\eta)^2} \mid \mathbf{x}, 1\right\}}{\widehat{\mathbb{E}}\left\{\frac{\varphi(\eta)-1}{\varphi(\eta)^2} \mid \mathbf{x}, 1\right\}} \right] = \mathbb{P}_n\left[ \frac{\varphi(\widehat{\eta}_{\text{eff}}) - a}{\varphi(\widehat{\eta}_{\text{eff}})} \frac{\mathbb{E}\left\{\frac{\dot{\varphi}(\eta)}{\varphi(\eta)^2} \mid \mathbf{x}, 1\right\}}{\mathbb{E}\left\{\frac{\varphi(\eta)-1}{\varphi(\eta)^2} \mid \mathbf{x}, 1\right\}} \right] + o_p(n^{-1/2}).
$$

By taking Taylor expansion, we have

$$
\begin{aligned}
&\mathbb{P}_n\left[ \frac{\varphi(\widehat{\eta}_{\text{eff}}) - a}{\varphi(\widehat{\eta}_{\text{eff}})} \frac{\mathbb{E}\left\{\frac{\dot{\varphi}(\eta)}{\varphi(\eta)^2} \mid \mathbf{x}, 1\right\}}{\mathbb{E}\left\{\frac{\varphi(\eta)-1}{\varphi(\eta)^2} \mid \mathbf{x}, 1\right\}} \right] \\
&= \mathbb{P}_n(S_{\eta,\text{eff}}) + \mathbb{P}\left[ \frac{a\dot{\varphi}(\eta)}{\varphi^2(\eta)} \frac{\mathbb{E}\left\{\frac{\dot{\varphi}(\eta)}{\varphi(\eta)^2} \mid \mathbf{x}, 1\right\}}{\mathbb{E}\left\{\frac{\varphi(\eta)-1}{\varphi(\eta)^2} \mid \mathbf{x}, 1\right\}} \right]^T (\widehat{\eta} - \eta) + o_p(n^{-1/2}) \\
&= \mathbb{P}_n(S_{\eta,\text{eff}}) - \text{Var}(S_{\eta,\text{eff}})(\widehat{\eta} - \eta) + o_p(n^{-1/2}).
\end{aligned}
\tag{19}
$$

By Assumption 5.1 and the empirical process theory, we have

$$
\begin{aligned}
\widehat{V}_{\text{eff}}(\pi) =& \mathbb{P}_n\left( \pi(\mathbf{x})\left[ \frac{a}{\varphi(\widehat{\eta}_{\text{eff}})} y + \left\{ 1 - \frac{a}{\varphi(\widehat{\eta}_{\text{eff}})} \right\} \frac{\mathbb{E}\left\{\frac{1-\varphi(\eta)}{\varphi(\eta)^2} Y \mid \mathbf{x}, 1\right\}}{\mathbb{E}\left\{\frac{1-\varphi(\eta)}{\varphi(\eta)^2} \mid \mathbf{x}, 1\right\}} \right] \right) \\
&+ \mathbb{P}\left[ \left\{ 1 - \frac{a}{\varphi(\widehat{\eta}_{\text{eff}})} \right\} \frac{\widehat{\mathbb{E}}\left\{\frac{1-\varphi(\eta)}{\varphi(\eta)^2} Y \mid \mathbf{x}, 1\right\}}{\widehat{\mathbb{E}}\left\{\frac{1-\varphi(\eta)}{\varphi(\eta)^2} \mid \mathbf{x}, 1\right\}} \right] - \mathbb{P}\left[ \left\{ 1 - \frac{a}{\varphi(\widehat{\eta}_{\text{eff}})} \right\} \frac{\mathbb{E}\left\{\frac{1-\varphi(\eta)}{\varphi(\eta)^2} Y \mid \mathbf{x}, 1\right\}}{\mathbb{E}\left\{\frac{1-\varphi(\eta)}{\varphi(\eta)^2} \mid \mathbf{x}, 1\right\}} \right] + o_p(n^{-1/2}).
\end{aligned}
\tag{20}
$$

For the ease of exposition, let $\mathbb{E}_3 = \mathbb{E}\left\{ \frac{1-\varphi(\eta)Y}{\varphi(\eta)^2} \mid \mathbf{x}, 1 \right\}$. By Assumptions 5.1, we have

$$
\begin{aligned}
& \left| \mathbb{P}\left\{ -\frac{\varphi(\widehat{\eta}_{\mathrm{eff}}) - a}{\varphi(\widehat{\eta}_{\mathrm{eff}})} \frac{\widehat{\mathbb{E}}_3}{\widehat{\mathbb{E}}_2} \right\} + \mathbb{P}\left\{ \frac{\varphi(\widehat{\eta}_{\mathrm{eff}}) - a}{\varphi(\widehat{\eta}_{\mathrm{eff}})} \frac{\mathbb{E}_3}{\mathbb{E}_2} \right\} \right| \\
={} & \left| \mathbb{P}\left[ \frac{\varphi(\widehat{\eta}_{\mathrm{eff}}) - a}{\varphi(\widehat{\eta}_{\mathrm{eff}})} \left\{ -\frac{\widehat{\mathbb{E}}_3}{\widehat{\mathbb{E}}_2} + \frac{\mathbb{E}_3}{\mathbb{E}_2} \right\} \right] \right| \\
={} & \left| \mathbb{P}\left[ \frac{\varphi(\widehat{\eta}_{\mathrm{eff}}) - a}{\varphi(\widehat{\eta}_{\mathrm{eff}})} \left\{ -\frac{\widehat{\mathbb{E}}_3}{\widehat{\mathbb{E}}_2} + \frac{\mathbb{E}_3}{\widehat{\mathbb{E}}_2} - \frac{\mathbb{E}_3}{\widehat{\mathbb{E}}_2} + \frac{\mathbb{E}_3}{\mathbb{E}_2} \right\} \right] \right| \\
={} & \left| \mathbb{P}\left[ \frac{\varphi(\widehat{\eta}_{\mathrm{eff}}) - a}{\varphi(\widehat{\eta}_{\mathrm{eff}})} \left\{ \frac{\mathbb{E}_3 - \widehat{\mathbb{E}}_3}{\widehat{\mathbb{E}}_2} + \frac{\mathbb{E}_3(\widehat{\mathbb{E}}_2 - \mathbb{E}_2)}{\mathbb{E}_2 \widehat{\mathbb{E}}_2} \right\} \right] \right| \\
\leq{} & O_p(n^{-1/2}) \times o_p(1) \\
={} & o_p(n^{-1/2}).
\end{aligned}
\tag{21}
$$

By Equations (20) and (21), we have

$$
\widehat{V}_{\mathrm{eff}}(\pi) = \mathbb{P}_n \left( \pi(\mathbf{x}) \left[ \frac{a}{\varphi(\widehat{\eta}_{\mathrm{eff}})} y + \left\{ 1 - \frac{a}{\varphi(\widehat{\eta}_{\mathrm{eff}})} \right\} \frac{\mathbb{E}\left\{ \frac{1-\varphi(\eta)}{\varphi(\eta)^2} Y \mid \mathbf{x}, 1 \right\}}{\mathbb{E}\left\{ \frac{1-\varphi(\eta)}{\varphi(\eta)^2} \mid \mathbf{x}, 1 \right\}} \right] \right) + o_p(n^{-1/2}).
$$

By taking Taylor expansion, we have

$$
\begin{aligned}
\widehat{V}_{\mathrm{eff}}(\pi) ={} & \mathbb{P}_n \left( \pi(\mathbf{x}) \left[ \frac{a}{\varphi(\eta)} y + \left\{ 1 - \frac{a}{\varphi(\eta)} \right\} \frac{\mathbb{E}\left\{ \frac{1-\varphi(\eta)}{\varphi(\eta)^2} Y \mid \mathbf{x}, 1 \right\}}{\mathbb{E}\left\{ \frac{1-\varphi(\eta)}{\varphi(\eta)^2} \mid \mathbf{x}, 1 \right\}} \right] \right) \\
& + \mathbb{P} \left( \pi(\mathbf{x}) \left[ -\frac{a\dot\varphi(\eta)}{\varphi^2(\eta)} y + \frac{a\dot\varphi(\eta)}{\varphi^2(\eta)} \frac{\mathbb{E}\left\{ \frac{1-\varphi(\eta)}{\varphi(\eta)^2} Y \mid \mathbf{x}, 1 \right\}}{\mathbb{E}\left\{ \frac{1-\varphi(\eta)}{\varphi(\eta)^2} \mid \mathbf{x}, 1 \right\}} \right] \right)^T (\widehat{\eta} - \eta) + o_p(n^{-1/2}).
\end{aligned}
\tag{22}
$$

By Equations (19) and (22), we have

$$
\begin{aligned}
& \widehat{V}_{\mathrm{eff}}(\pi) - V_1(\pi) \\
={} & \mathbb{P}_n \left( \pi(\mathbf{x}) \left[ \frac{a}{\varphi(\eta)} y + \left\{ 1 - \frac{a}{\varphi(\eta)} \right\} \frac{\mathbb{E}\left\{ \frac{1-\varphi(\eta)}{\varphi(\eta)^2} Y \mid \mathbf{x}, 1 \right\}}{\mathbb{E}\left\{ \frac{1-\varphi(\eta)}{\varphi(\eta)^2} \mid \mathbf{x}, 1 \right\}} \right] \right) \\
& + \mathbb{P} \left( \pi(\mathbf{x}) \left[ -\frac{a\dot\varphi(\eta)}{\varphi^2(\eta)} y + \frac{a\dot\varphi(\eta)}{\varphi^2(\eta)} \frac{\mathbb{E}\left\{ \frac{1-\varphi(\eta)}{\varphi(\eta)^2} Y \mid \mathbf{x}, 1 \right\}}{\mathbb{E}\left\{ \frac{1-\varphi(\eta)}{\varphi(\eta)^2} \mid \mathbf{x}, 1 \right\}} \right] \right)^T \{\mathrm{Var}(S_{\eta,\mathrm{eff}})\}^{-1} \mathbb{P}_n(S_{\eta,\mathrm{eff}}) - V_1(\pi) + o_p(n^{-1/2}) \\
={} & \mathbb{P}_n \left( \pi(\mathbf{x}) \left[ \frac{a}{\varphi(\eta)} y + \left\{ 1 - \frac{a}{\varphi(\eta)} \right\} \frac{\mathbb{E}\left\{ \frac{1-\varphi(\eta)}{\varphi(\eta)^2} Y \mid \mathbf{x}, 1 \right\}}{\mathbb{E}\left\{ \frac{1-\varphi(\eta)}{\varphi(\eta)^2} \mid \mathbf{x}, 1 \right\}} \right] \right) + \boldsymbol{D}^T \mathbb{P}_n(S_{\eta,\mathrm{eff}}) - V_1(\pi) + o_p(n^{-1/2}) \\
={} & \mathbb{P}_n \left( \pi(\mathbf{x}) \left[ \frac{a}{\varphi(\eta)} y + \left\{ 1 - \frac{a}{\varphi(\eta)} \right\} \frac{\mathbb{E}\left\{ \frac{1-\varphi(\eta)}{\varphi(\eta)^2} Y \mid \mathbf{x}, 1 \right\}}{\mathbb{E}\left\{ \frac{1-\varphi(\eta)}{\varphi(\eta)^2} \mid \mathbf{x}, 1 \right\}} \right] + \boldsymbol{D}^T S_{\eta,\mathrm{eff}} - V_1(\pi) \right) + o_p(n^{-1/2}) \\
={} & \mathbb{P}_n \{\phi_{\mathrm{eff}}(\pi)\} + o_p(n^{-1/2}).
\end{aligned}
$$

This completes the proof. $\qquad\square$

A.6 PROOF OF PROPOSITION 5.3

$$\arg\max_{\pi\in\Pi} \widehat{V}_{\text{eff}}(\pi)$$

$$= \arg\max_{\pi\in\Pi} \sum_{i=1}^{n} \pi(\mathbf{x}_i)\widehat{\psi}(\mathbf{x}_i, y_i, a_i)$$

$$= \arg\max_{\pi\in\Pi} \sum_{i=1}^{n} \pi(\mathbf{x}_i)|\widehat{\psi}(\mathbf{x}_i, y_i, a_i)|[\mathbb{I}\{\widehat{\psi}(\mathbf{x}_i, y_i, a_i) > 0\} - \mathbb{I}\{\widehat{\psi}(\mathbf{x}_i, y_i, a_i) \le 0\}]$$

$$= \arg\max_{\pi\in\Pi} \sum_{i=1}^{n} |\widehat{\psi}(\mathbf{x}_i, y_i, a_i)|\mathbb{I}\{\widehat{\psi}(\mathbf{x}_i, y_i, a_i) > 0\}$$
$$- |\widehat{\psi}(\mathbf{x}_i, y_i, a_i)|[\{1 - \pi(\mathbf{x}_i)\}\mathbb{I}\{\widehat{\psi}(\mathbf{x}_i, y_i, a_i) > 0\} + \pi(\mathbf{x}_i)\mathbb{I}\{\widehat{\psi}(\mathbf{x}_i, y_i, a_i) \le 0\}]$$

$$= \arg\max_{\pi\in\Pi} \sum_{i=1}^{n} |\widehat{\psi}(\mathbf{x}_i, y_i, a_i)|\mathbb{I}\{\widehat{\psi}(\mathbf{x}_i, y_i, a_i) > 0\}$$
$$- |\widehat{\psi}(\mathbf{x}_i, y_i, a_i)|[\pi(\mathbf{x}_i) + \mathbb{I}\{\widehat{\psi}(\mathbf{x}_i, y_i, a_i) > 0\} - 2\pi(\mathbf{x}_i)\mathbb{I}\{\widehat{\psi}(\mathbf{x}_i, y_i, a_i) > 0\}]$$

$$= \arg\max_{\pi\in\Pi} \sum_{i=1}^{n} |\widehat{\psi}(\mathbf{x}_i, y_i, a_i)|\mathbb{I}\{\widehat{\psi}(\mathbf{x}_i, y_i, a_i) > 0\}$$
$$- |\widehat{\psi}(\mathbf{x}_i, y_i, a_i)|[\pi^2(\mathbf{x}) + \mathbb{I}^2\{\widehat{\psi}(\mathbf{x}_i, y_i, a_i) > 0\} - 2\pi(\mathbf{x}_i)\mathbb{I}\{\widehat{\psi}(\mathbf{x}_i, y_i, a_i) > 0\}]$$

$$= \arg\max_{\pi\in\Pi} \sum_{i=1}^{n} |\widehat{\psi}(\mathbf{x}_i, y_i, a_i)|\mathbb{I}\{\widehat{\psi}(\mathbf{x}_i, y_i, a_i) > 0\} - |\widehat{\psi}(\mathbf{x}_i, y_i, a_i)|[\pi(\mathbf{x}_i) - \mathbb{I}\{\widehat{\psi}(\mathbf{x}_i, y_i, a_i) > 0\}]^2$$

$$= \arg\max_{\pi\in\Pi} \sum_{i=1}^{n} -|\widehat{\psi}(\mathbf{x}_i, y_i, a_i)|[\pi(\mathbf{x}_i) - \mathbb{I}\{\widehat{\psi}(\mathbf{x}_i, y_i, a_i) > 0\}]^2$$

$$= \arg\min_{\pi\in\Pi} \sum_{i=1}^{n} |\widehat{\psi}(\mathbf{x}_i, y_i, a_i)|[\pi(\mathbf{x}_i) - \mathbb{I}\{\widehat{\psi}(\mathbf{x}_i, y_i, a_i) > 0\}]^2$$

$$= \arg\min_{\pi\in\Pi} \sum_{i=1}^{n} |\widehat{\psi}(\mathbf{x}_i, y_i, a_i)|\mathbb{I}[\pi(\mathbf{x}_i) \ne \mathbb{I}\{\widehat{\psi}(\mathbf{x}_i, y_i, a_i) > 0\}].$$

Therefore, the decision learning is equivalent to a weighted classification problem, where for subject $i$ with features $\mathbf{x}_i$, the true label is $\mathbb{I}\{\widehat{\psi}(\mathbf{x}_i, y_i, a_i) > 0\}$ and the sample weight is $|\widehat{\psi}(\mathbf{x}_i, y_i, a_i)|$.

$\square$

