# OpenReview forum: "Efficient Causal Decision Making with One-sided Feedback"
_ICLR.cc/2025/Conference — ICLR 2025 Poster_

### Official Review · Reviewer_h5vG · 2024-10-28

**Soundness:** 2
**Presentation:** 3
**Contribution:** 2
**Rating:** 6
**Confidence:** 2

**Summary:**

The authors study a class of decision-making problems with one-sided feedback, where outcomes are only observable for specific actions. They introduce a value function to evaluate decision rules that address the issue of undefined counterfactual outcomes. Without assuming no unmeasured confounders, they establish the identification of the value function using shadow variables. Furthermore, leveraging semiparametric theory, they derive the efficiency bound for the proposed value function and develop efficient methods for decision evaluation and learning. Numerical experiments and a real-world data application demonstrate the empirical performance of proposed methods.

**Strengths:**

The paper is well-written. The paper establishes rigorous identification results and semiparametric theory.

**Weaknesses:**

Please refer to Questions.

**Questions:**

It seems that the framework essentially treats all $Y(0)$ as 0. Then is the task just to predict the sign of $Y(1)$?

---

> ### Author Response · Authors · 2024-11-15
> **Response to Reviewer h5vG**
>
> Thanks very much for the insightful comment.
> Yes, $Y(0)$ is essentially treated as 0 as this is the natural way to define $Y(0)$ in the one-sided feedback problem such as bank loan. However, the decision making task with one-sided feedback is not just to predict the sign of $Y(1)$.
>
> First, in the **decision evaluation** task, we aim to estimate the precise quantity of the value function $V_1(\pi)$. To this end, we have developed two different estimation strategies for binary and continuous outcomes.
>
> Second, in the **decision learning** task, we demonstrate that the problem can be reformulated as a **weighted classification problem** (Proposition 5.3). Here, the label is $L={\mathbb{I}}\\{\widehat{\psi}(X,Y,A)>0\\}$ is based on whether an estimated value
> $\widehat{\psi}(X,Y,A)$, which approximates $\mathbb{E}\\{Y(1)\mid X\\}$, is positive. Additionally, we also need the estimated **weights**  $W = |\widehat{\psi}(X,Y,A)|$, which requires estimating the quantity of $\mathbb{E}\\{Y(1)\mid X\\}$. This task is particularly challenging, as we can only observe  $Y(1)$ for individuals who received action $A=1$, and without the no unmeasured confounders assumption, we do not have that $\mathbb{E}\\{Y(1)\mid X,A=1\\}=\mathbb{E}\\{Y(1)\mid X,A=0\\}$. To address this, we establish identification results and develop estimation strategies using shadow variables.

---

> > ### Author Response · Authors · 2024-11-21
> >
> > Thank you again for your feedback on our submission. We greatly value your insights and have addressed your comments in our rebuttal. We would greatly appreciate your response as the discussion period deadline is approaching. If there are any additional concerns or points needing clarification, we’d be happy to provide further details to ensure your questions are resolved.
> >
> > Thank you for your time and support in the review process!

---

> > > ### Comment · Reviewer_h5vG · 2024-11-26
> > >
> > > Thank you for the clarifications. It addressed my comments and I will update my score accordingly.

---

### Official Review · Reviewer_oarq · 2024-10-29

**Soundness:** 3
**Presentation:** 3
**Contribution:** 3
**Rating:** 8
**Confidence:** 2

**Summary:**

The paper addresses decision-making scenarios where only specific actions yield observable outcomes. Since counterfactuals are undefined in this setting, a new method beyond conventional causal inference approaches is needed. The paper proposes a novel value function to evaluate decision rules in one-sided feedback contexts without assuming no unmeasured confounders. An efficiency bound, $\Gamma(\pi)$, is provided for the value function. The paper also introduces efficient estimators for decision evaluation and learning. Through simulations and applications on real-world data, including a bank loan dataset, the paper demonstrates that the proposed method surpasses traditional approaches in robustness and efficiency for evaluating and learning decision rules.

**Strengths:**

Please find the strengths below:
1. The paper studies an important problem in practice. In applications like credit card applications and bank loans, the outcomes may not be observable when certain actions are taken.
2. The algorithm for learning and evaluating the causal relationship is theoretically effective. The use of shadow variables is innovative and removes the requirement of the NUC assumption.
3. The paper provides both numerical and practical experiments. The experiments show the efficiency of the proposed estimators for the one-sided feedback setting.
4. The method in the paper appears flexible and adaptable to me, allowing it to be extended to a variety of settings with partial feedback.

**Weaknesses:**

Please find the weaknesses below:
1. The paper only considers binary actions, i.e., $A=0,1$, which is relatively limited. Given that this is an initial work on partial feedback learning, this limitation is acceptable to me.
2. I did not find any discussion on the optimality of the semiparametric efficiency bound $\Gamma(\pi)$.
3. The numerical experiment is based on a specific setting, which is not general enough (both the action distribution and the outcome function are fixed). The experiment on the bank loan dataset does not show a very notable improvement of the proposed method over IPW-eff, compared to the results in the numerical experiment.

**Questions:**

Refer to the weaknesses section above, and find the additional questions below:
1. Is it possible to extend the work to non-binary action settings and more general partial feedback causality settings?
2. Including a wider range of settings in the numerical experiments would make the results more persuasive.
3. The theoretical analysis considers binary and continuous outcome variables. Does the theoretical analysis also apply to categorical outcomes?

**Details Of Ethics Concerns:**

I did not identify any ethical concerns.

---

> ### Author Response · Authors · 2024-11-20
> **Response to Reviewer oarq (part I)**
>
> We sincerely appreciate your insightful and constructive comments. Below, we have provided our detailed, point-by-point responses.
>
> **Extension to non-binary action settings**: Thanks for the insightful comment. Yes, the work can be naturally generalizes to multi-action settings. Consider the practical example of advertising: an advertiser observes user engagement (such as clicks) only when an ad is shown to the user. When no ad is displayed, the advertiser receives no information about whether the user might have clicked or engaged with the ads.
>
> Suppose there are $K$ ads in total. Each ad
> $k\in\\{1, \dots, K\\}$ can be considered as an action, while the "no show" scenario can be treated as action
> $A=0$. In this case, the performance of ads (e.g., click-through rates) is only observed when an action is taken from
> $\\{1, \dots, K\\}$. No feedback is received when no ad is displayed ($A=0$). We can define the potential outcome for action $k$ as $Y(k)$, the decision rule as $\pi(x,k)=P(A=k\mid X=x)$ and the corresponding value function as $V(\pi)=\mathbb{E}\\{\sum_{k=1}^KY(k)\pi(X,k)\\}$. The identification and estimation strategies can be similarly established using shadow variables.
>
> **Optimality of the semiparametric efficiency bound $\Gamma(\pi)$**: The semiparametric efficiency bound is defined as the supremum of the Cramer-Rao lower bounds for all parametric submodels. This bound describes how difficult it is to estimate the target estimand, namely the value function in our context, under the general semiparametric framework.  A value estimator that achieves the semiparametric efficiency bound $\Gamma(\pi)$ is the most efficient estimator (with minimum asymptotic variance) within the class of semiparametric models.
>
> **Wider range of numerical experiments**: In the real data application, we observe that the means of the proposed method’s values are higher than those of IPW-eff, while the variation is much smaller. The reason this improvement is less pronounced compared to synthetic scenarios is that the scale of the binary outcome is much smaller (the outcome
> is defined as 1 if the loan is repaid and -1 if the applicant defaults).
>
> For the synthetic scenarios, we have conducted additional experiments with more complex outcome functions and treatment assignment mechanisms. Specifically, in Case 1, the potential outcome is generated by $Y(1) = 6X_1 - 3X_1^2 -3X_2 + 6X_3^2 + 3X_1X_2+3X_1X_3+1.5X_1X_2 +\epsilon$,
> where $\epsilon$ is generated from a normal distribution with mean 0 and standard deviation 0.5. The action $A$ is generated from $A\sim \text{Bernoulli}\\{\varphi(X,Y(1))\\}$, and $\text{logit}\\{
> \varphi(X,Y(1))\\} =1/[1+\exp\{0.5-X_1-X_1X_2-0.1Y(1)\}]$. In Case 2, the potential outcome $Y(1)$ follows a Bernoulli distribution with probability of success $1/\{1+\exp(2X_1+X_2+X_3+X_1X_2+X_1X_3+2X_2X_3)\}$. The action $A$ is generated from $A\sim \text{Bernoulli} \\{\varphi(X,Y(1))\\}$, and $\text{logit}\\{
> \varphi(X,Y(1))\\} = 1/[1+\exp\{-0.5-X_1+0.5X_1X_2-0.5Y(1)\}]$. The other settings are the same. We present decision evaluation results in Table 1 and Table 2.
>
> **Table 1**: Simulation results for case 1: (a) $0.0\pi_d+1.0\pi_u$, (b) $0.3\pi_d+0.7\pi_u$, (c)  $0.6\pi_d+0.4\pi_u$.
> |                   | |    (a)  |      |  | (b)   |      |   | (c)     |       |
> |-------------------|--------|--------|------------|--------|--------|------------|--------|--------|------------|
> |                   | RMSE   | SD     | Bias       | RMSE   | SD     | Bias       | RMSE   | SD     | Bias       |
> | **n=1000**        |        |        |            |        |        |            |        |        |            |
> | $\widehat{V}_{\rm{eff}}$         | 0.3383 | **0.3298** | 0.0754    | 0.3771 | **0.3734** | 0.0525    | 0.4374 | **0.4364** | 0.0297    |
> | $\widehat{V}_{\rm{IPW-naive}}$  | 1.5189 | 1.2341 | 0.8855    | 1.8413 | 1.5066 | 1.0585    | 2.1685 | 1.7848 | 1.2315    |
> | $\widehat{V}_{\rm{IPW-eff}}$    | 0.4661 | 0.4511 | 0.1172    | 0.5822 | 0.5751 | 0.0905    | 0.7149 | 0.7121 | 0.0638    |
> | $\widehat{V}_{\rm{DR}}$         | 0.7695 | 0.2077 | 0.7409    | 0.8151 | 0.2531 | 0.7748    | 0.8630 | 0.3010 | 0.8088    |
> |-------------------|--------|--------|------------|--------|--------|------------|--------|--------|------------|
> | **n=2000**        |        |        |            |        |        |            |        |        |            |
> | $\widehat{V}_{\rm{eff}}$         | 0.2248 | **0.2147** | 0.0666    | 0.2336 | **0.2270** | 0.0551    | 0.2577 | **0.2540** | 0.0437    |
> | $\widehat{V}_{\rm{IPW-naive}}$  | 1.3439 | 1.1197 | 0.7432    | 1.6188 | 1.3524 | 0.8895    | 1.8979 | 1.5902 | 1.0359    |
> | $\widehat{V}_{\rm{IPW-eff}}$    | 0.2937 | 0.2891 | 0.0520    | 0.2863 | 0.2848 | 0.0290    | 0.2961 | 0.2961 | 0.0060    |
> | $\widehat{V}_{\rm{DR}}$         | 0.6432 | 0.1397 | 0.6279    | 0.6698 | 0.1643 | 0.6493    | 0.6975 | 0.1912 | 0.6708    |

---

> > ### Author Response · Authors · 2024-11-20
> > **Response to Reviewer oarq (part II)**
> >
> > Cont’d
> >
> > **Table 2**: Simulation results for case 2. (a) $0.0\pi_d+1.0\pi_u$, (b) $0.4\pi_d+0.6\pi_u$, (c)  $0.7\pi_d+0.3\pi_u$.
> > |                   | |    (a)  |      |  | (b)   |      |   | (c)     |       |
> > |-------------------|--------|--------|------------|--------|--------|------------|--------|--------|------------|
> > |                   | RMSE   | SD     | Bias       | RMSE   | SD     | Bias       | RMSE   | SD     | Bias       |
> > | **n = 1000**              |           |          |           |           |          |           |           |          |           |
> > | $\widehat{V}_{\rm{eff}}$   | 0.0149    | **0.0148**   | -0.0016   | 0.0173    | **0.0171**   | -0.0026   | 0.0199    | **0.0196**   | -0.0033   |
> > | $\widehat{V}_{\rm{IPW-naive}}$ | 0.0168    | 0.0167   | 0.0019    | 0.0192    | 0.0192   | 0.0011    | 0.0220    | 0.0219   | 0.0005    |
> > | $\widehat{V}_{\rm{IPW-eff}}$ | 0.0153    | 0.0152   | -0.0016   | 0.0180    | 0.0178   | -0.0024   | 0.0208    | 0.0206   | -0.0030   |
> > | $\widehat{V}_{\rm{DR}}$    | 0.0130    | 0.0081   | 0.0102    | 0.0143    | 0.0099   | 0.0102    | 0.0163    | 0.0126   | 0.0103    |
> > | **n = 2000**              |           |          |           |           |          |           |           |          |           |
> > | $\widehat{V}_{\rm{eff}}$   | 0.0100    | **0.0100**   | 0.0005    | 0.0114    | **0.0114**   | 0.0002    | 0.0130    | **0.0130**   | -0.0001   |
> > | $\widehat{V}_{\rm{IPW-naive}}$ | 0.0130    | 0.0128   | 0.0020    | 0.0144    | 0.0143   | 0.0016    | 0.0161    | 0.0161   | 0.0013    |
> > | $\widehat{V}_{\rm{IPW-eff}}$ | 0.0102    | 0.0102   | 0.0005    | 0.0118    | 0.0118   | 0.0003    | 0.0136    | 0.0136   | 0.0001    |
> > | $\widehat{V}_{\rm{DR}}$    | 0.0119    | 0.0059   | 0.0104    | 0.0131    | 0.0073   | 0.0109    | 0.0145    | 0.0091   | 0.0112    |
> >
> >  We observe similar patterns to the previous results: $\widehat{V}\_{\rm{eff}}$,  and $\widehat{V}\_{\rm{IPW-eff}}$ are nearly unbiased with sample size $n=1000,2000$. However, $\widehat{V}\_{\rm{DR}}$ has a significantly larger bias when compared to other estimators. This is because the NUC assumption is violated in this setting.  Among three consistent estimators $\widehat{V}\_{\rm{eff}}$,$\widehat{V}\_{\rm{IPW-naive}}$, and $\widehat{V}\_{\rm{IPW-eff}}$, $\widehat{V}_{\rm{eff}}$ has the smallest standard deviation and RMSE.  The decision learning results are also similar to those of previous experiments. We will include plots in the revised version.
> >
> > **Categorical outcomes**: Yes, our theoretical analysis extends to categorical outcomes as well. We recommend using a similar estimation strategy to that for binary outcomes. Say $Y\in\\{1,\dots, K\\}$, we can specify a model for $P(Y=k\mid X,A=1)$ for $k\in \\{1,\dots, K\\}$, and we denote its estimator as  $\widehat{P}(Y=k\mid X,A=1)$. The conditional expectations in (8) can be estimated by
> > $
> > \widehat{\mathbb{E}}\left\\{\frac{\dot{\varphi}(\eta)}{\varphi(\eta)^2}\mid X, A=1\right\\}
> > = \sum_{k=1}^K\frac{1}{\varphi(U,k;\eta)^2}\frac{\partial \varphi(U,k;\eta)}{\partial \eta}\widehat{P}(Y=k\mid X,A=1),
> > $
> > and $\widehat{\mathbb{E}}\left\\{\frac{\varphi(\eta)-1}{\varphi(\eta)^2}\mid X, A=1\right\\} = \sum_{k=1}^K\frac{\varphi(U,k;\eta)-1}{\varphi(U,k;\eta)^2}\widehat{P}(Y=k\mid X,A=1).$
> > Thus we can get the efficient estimator $\widehat{\eta}\_{\rm{eff}}$ by solving (8). Next, the conditional expectations in (6) can be estimated by
> > $\widehat{\mathbb{E}}\left\\{\frac{1-\varphi(\eta)}{\varphi(\eta)^2}Y\mid X,A=1\right\\}
> > = \sum_{k=1}^Kk\frac{1-\varphi(U,k;\widehat{\eta}\_{\rm{eff}})}{\varphi(U,k;\widehat{\eta}\_{\rm{eff}})^2}\widehat{P}(Y=1\mid X,A=1)
> > $, and
> > $\widehat{\mathbb{E}}\left\\{\frac{1-\varphi(\eta)}{\varphi(\eta)^2}\mid X,A=1\right\\} =\sum_{k=1}^K\frac{1-\varphi(U,k;\widehat{\eta}\_{\rm{eff}})}{\varphi(U,k;\widehat{\eta}\_{\rm{eff}})^2}\widehat{P}(Y=k \mid X,A=1).
> > $ By plugging the estimators $\widehat{\eta}\_{\rm{eff}}$, $\widehat{\mathbb{E}}\left\\{\frac{1-\varphi(\eta)}{\varphi(\eta)^2}Y\mid X,A=1\right\\}$,
> > and $\widehat{\mathbb{E}}\left\\{\frac{1-\varphi(\eta)}{\varphi(\eta)^2}\mid X,A=1\right\\}$ into (6), we obtain the value estimator and denote it as $\widehat{V}\_{\rm{eff}}(\pi)$. And Theorem 5.2 applies to this scenario.

---

> > > ### Comment · Reviewer_oarq · 2024-11-21
> > >
> > > Thank you for providing the additional experiments and clarifications. I believe this is a strong paper, and I will maintain my accept score.

---

### Official Review · Reviewer_d6pz · 2024-11-01

**Soundness:** 2
**Presentation:** 3
**Contribution:** 2
**Rating:** 6
**Confidence:** 2

**Summary:**

The paper defines a new value function for the problem of 'one-sided feedback' in causal inference for decision making.

It provides identification results, using the so called 'shadow variables'.

The authors also provide efficiency bounds using semi-parametric.

Finally, they also provide empirical evidence for their new method.

**Strengths:**

The paper is strong in exposition of its core idea, methods, experiments and results.

It seems technically sound.

**Weaknesses:**

I struggle to accept the core assumption made in line 157:

"On the other hand, if π(X) = 0, indicating loan rejection, the bank neither earns nor loses any money. Therefore, the newly defined value function V1(π) quantifies the expected monetary outcome for the bank when implementing decision rule π for loan approvals"

I want to argue that the bank in fact does lose money when it rejects a loan. Some of the loans, might have in fact not defaulted and therefore produced a profit for the bank. This concept is known to me as "opportunity cost", sometimes defined as "the loss of potential gain from other alternatives when one alternative is chosen".

In this case, it'd be the loss of the potential gain from giving out the loan, when choosing to reject the loan applicant.

There seems to be no discussion of whether this new definition of the value function makes conceptually sense. It is assumed to be valid, and technical challenges are solved (probably successfully).

**Questions:**

Can you help me understand why your new value function is of practical value and relevance?

- Is the concept of 'opportunity cost' relevant? Why or why not?
- Is there a practical background for this new value function? is it already being used in decision making elsewhere?
- What was the inspiration for this value function? Is it simply the observation that we can't do anything otherwise or is there a more fundamental observation from e.g. banking where this function is being actively used?

If it's just a technical convenience, but no practical relevance, than the contribution of this paper is hard to justify without a stronger discussion of why a user should accept this value function.

If there is more to it than just a technical connivence, than I am very keen to learn about that!

Thanks!

---

> ### Author Response · Authors · 2024-11-15
>
> Thanks very much for the insightful comment.
>
> Yes, opportunity cost is exactly reflected in our new value function. The value function $V_1(\pi)= \mathbb{E}\\{Y(1)\pi(X)\\}$ is defined to capture the outcomes of decisions under one-sided feedback, reflecting **missed potential** in cases where an action is not taken. Specifically, we define the potential outcome $Y(1)$ as **the outcome if an individual were approved**.
> If $Y(1)$ is positive, it indicates a profit opportunity, for instance, in a banking context where approving a loan could lead to financial gains. When the decision is to approve ($\pi(X)=1$), the profit is $Y(1)\pi(X)=Y(1)\times 1 = Y(1)$. However, if the loan is rejected ($\pi(X)=0$), the profit is $Y(1)\times 0 =0$. Thus, the opportunity loss is measured by $Y(1)-0$ under the rejection decision, which is exactly reflected in the value function because the value function becomes smaller with such a rejection decision. Therefore, the value function $V_1(\pi)= \mathbb{E}\\{Y(1)\pi(X)\\}$ directly incorporates this concept of opportunity cost, quantifying the outcome of both the action taken and the loss incurred from not taking it.
>
> Beyond the banking example, this framework is relevant to many decision-making scenarios with one-sided feedback, such as hiring, admission and advertising. In the hiring/admission example, one-sided feedback arises because a decision-maker only observes candidates' actual performance if they decide to hire/admit them. If a candidate is rejected, the decision-maker receives no feedback $Y(1)$ about how that individual might have performed in the role. This one-sided feedback reflects opportunity cost in hiring/admission decisions, as the decision-maker misses good performance outcomes if strong performers are declined.
> In advertising, an advertiser only observes user engagement (such as clicks) if the ad is shown to the user. When the ad is not displayed, the advertiser receives no information $Y(1)$ about whether the user might have clicked on it or engaged with it. Thus, the outcome—whether the ad would have been effective—is only observed if the action (showing the ad) is taken. This one-sided feedback reflects opportunity cost as well: if the ad is not shown, the advertiser misses a potential click or engagement that could have led to a conversion or sale.
>
> The inspiration for this value function stems from a need to reflect real-world decision-making processes with one-sided feedback. This is not merely a technical convenience; rather, it is based on a fundamental need to address opportunity costs explicitly. Hence, this value function is useful and interpretable in many practical applications where decisions must account for potential outcomes that are only observable under certain conditions.

---

> > ### Comment · Reviewer_d6pz · 2024-11-20
> > **Thanks for the clarifications**
> >
> > Can you provide some references from e.g. the economics literature that support your proposed value function and opportunity cost? I'd find it hard to believe that this has not been discussed there!
> >
> > I am not trying to devalue your contribution, just worried about other readers like myself looking for more (economic) decision making context and background, and I believe some references would greatly help with your exposition.
> >
> > Thank you very much in advance.

---

> > > ### Author Response · Authors · 2024-11-20
> > >
> > > Thank you very much for the suggestion! Below are some references that support the motivation of our work (some of which are already mentioned in the introduction and related work sections). We will expand the discussion of these works when presenting the motivation, as we believe this will enhance readers' understanding of our work's foundation and clarify why the newly defined value function is significant for causal decision-making in settings with one-sided feedback.
> > >
> > > 1. Harris, Keegan, Chara Podimata, and Steven Z. Wu. "Strategic apple tasting." Advances in Neural Information Processing Systems 36 (2023): 79918-79945.
> > >
> > > 2. Pacchiano, Aldo, et al. "Neural pseudo-label optimism for the bank loan problem." Advances in Neural Information Processing Systems 34 (2021): 6580-6593.
> > >
> > > 3. Lakkaraju, Himabindu, et al. "The selective labels problem: Evaluating algorithmic predictions in the presence of unobservables." Proceedings of the 23rd ACM SIGKDD International Conference on Knowledge Discovery and Data Mining. 2017.
> > >
> > > 4. Heinrich Jiang, Qijia Jiang, and Aldo Pacchiano. Learning the truth from only one side of the story. In International Conference on Artificial Intelligence and Statistics, pp. 2413–2421. PMLR, 2021.

---

### Official Review · Reviewer_Wzn4 · 2024-11-02

**Soundness:** 3
**Presentation:** 3
**Contribution:** 4
**Rating:** 6
**Confidence:** 3

**Summary:**

This paper addresses decision-making with one-sided feedback, where outcomes are observable only for chosen actions, such as approved loans. The authors introduce a new value function for evaluating decisions without assuming the no unmeasured confounders (NUC) condition. Using shadow variables, they establish identifiability and derive the efficient influence function (EIF) and the semiparametric efficiency bound of the value function. Motivated by the EIF, they develop two efficient estimators for the value function, applicable to binary and continuous outcomes, respectively. They propose a weighted, classification-based framework to learn the optimal decision rule. Empirical results validate the method's effectiveness.

**Strengths:**

The paper's main strength  is introducing a modified value function and establishing identifiability for this value function by leveraging so called shadow variables (SVs). The authors derive the efficient influence function (EIF) and the semiparametric efficiency bound of the value function. Additionally, in the case of continuous distributions, their proposed estimation strategy avoids estimating the density when the outcome is continuous, thus preventing instability. Furthermore, they demonstrate that their estimators are consistent and achieve the semiparametric efficiency bound under mild conditions for nuisance function approximation.

**Weaknesses:**

The assumption of knowing the shadow variables (SVs) may restrict the practical application of the proposed approach. Similarly, assumption 4.2 could also limit the applicability of the proposed methods. Although the authors cite some relevant papers for assumption 4.2, more discussion is needed to clarify how restrictive this assumption is.  The proposed methodology requires that conditional expectations \( P(Y | A = 1, X) \) be consistently estimated. This might be an issue particularly for high-dimensional discrete covariates, i.e., categorical variables that can take on a large number of distinct categories or levels.

**Questions:**

I have some questions/comments for the authors:

* The authors state that assumption 4.2 guarantees the uniqueness of $ \phi(u,y) $. Can the authors explain how this assumption guarantees this? Also, is it a sufficient condition for uniqueness, or could there be necessary or more relaxed conditions?

* Regarding assumption 4.2, it is worthwhile to discuss in the paper how restrictive this assumption is. Perhaps the authors could mention some cases where it does not hold.

* How do high-dimensional categorical covariates affect the estimation for the conditional expectations as required by assumption 5.1?

* The authors mentioned that the maximizer for the efficient estimator denoted by $ \hat{\pi} $ is a natural estimator for the true optimal decision rule $ \pi^* $, which is a maximizer of the true value function. Although in experiments, the authors show percentages of making correct decisions (PCD), can they provide any discussion on how far $ \hat{\pi} $ can be from the true optimal rule $ \pi^* $ in general?

---

> ### Author Response · Authors · 2024-11-20
> **Response to Reviewer Wzn4 (part I)**
>
> We sincerely appreciate your insightful and constructive comments. Below, we have provided our detailed, point-by-point responses.
>
> **Weaknesses**:
>
> **Shadow variable assumption**: Shadow variables (SVs) can be identified through two main approaches. **First**, we can leverage expert knowledge in specific domains. For instance, in fairness-oriented hiring or admissions scenarios, sensitive attributes such as age and gender, which are unrelated to action assignments, can serve as SVs. Similarly, in medical treatment contexts, when the intervention is conveniently deliverable, factors like a patient’s access to transportation or distance to a healthcare center, which are unrelated to treatment assignment, can be considered SVs. **Second**, when we do not have prior expert knowledge, we can still automatically generate representations from observed covariates that serve the role of SVs, as demonstrated in Li et al. (2024).
> Therefore, SVs can be either selected or generated in practical settings, reducing the reliance on restrictive prior assumptions.
>
> **Conditional completeness assumption (Assumption 4.2)**: Assumption 4.2 is necessary to guarantee the nonparametric identification of $\phi(u, y)$. However, it is not restrictive in many practical scenarios. As discussed in the paper, when $Y(1)$ is binary, we only require $Z$ to be either continuous or a categorical variable with more than two levels. For cases where $Y(1)$ is continuous, the assumption holds if $f(y \mid x, 1)$ belongs to certain common distributions, such as exponential families and location-scale families. Hu and Shiu (2018) provides some other sufficient conditions for this assumption in different applications. We will include more detailed discussion of this assumption in the revised version.
>
> Reference: Hu, Y. and Shiu, J.L., 2018. Nonparametric identification using instrumental variables: sufficient conditions for completeness. Econometric Theory, 34(3), pp.659-693.
>
> **Consistency of regression with high-dimensional discrete covariates**: Regression with high-dimensional discrete covariates is indeed challenging, as converting these variables into numerical forms (e.g., one-hot encoding) can significantly increase the feature space. This often results in a sparse design matrix, particularly when most data points belong to a small subset of levels, complicating computation and optimization.
>
> As discussed in the paper, our methodology supports the use of machine learning and deep learning models to fit the regression, allowing for various strategies to address these challenges:
>
> **Embeddings**: Use learned representations from neural networks to represent categorical variables in a dense, lower-dimensional space.
> **Level Clustering**: Cluster levels based on similarity (e.g., frequency or behavior) to reduce cardinality.
> **Tree-Based Models**: Employ models like random forests or gradient boosting, which can handle categorical variables natively without explicit encoding.
> **Regularization**: Apply techniques such as Lasso, Ridge, or group lasso to prevent overfitting.
> **Bayesian Hierarchical Models**: Pool information across levels, improving estimation for rare categories.
> We will include this discussion in the revised version of the paper to address these considerations in greater detail.

---

> ### Author Response · Authors · 2024-11-20
> **Response to Reviewer Wzn4 (part II)**
>
> We sincerely appreciate your insightful and constructive comments. Below, we have provided our detailed, point-by-point responses.
>
> **Questions**:
> > 1. The authors state that assumption 4.2 guarantees the uniqueness of $\phi(u,y)$. Can the authors explain how this assumption guarantees this? Also, is it a sufficient condition for uniqueness, or could there be necessary or more relaxed conditions?
>
> **Re**: The observed distribution $w(x)=P(A=1\mid X=x)$ has the following relationship with $\varphi(u,y)$:
> $$w^{-1}(x) = \int \frac{f(y|x,1)}{\varphi(u,y)}dy.$$
> Suppose there exist two different quantities $\varphi(u,y)$ and $\tilde{\varphi}(u,y)$ that satisfy the above equation. Then we have
> $$
> \int\frac{f(y|x,1)}{\varphi(u,y)}dy=\int\frac{f(y|x,1)}{\tilde{\varphi}(u,y)}dy.
> $$
> Let $h(u,y) = \frac{1}{\varphi(u,y)}- \frac{1}{\tilde{\varphi}(u,y)}$. Substituting this into the equation, we obtain:
> $$
> \int h(u,y)f(y|x,1)dy=0.
> $$
> By Assumption 4.2, it follows directly that $h(u,y)=\frac{1}{\varphi(u,y)}- \frac{1}{\tilde{\varphi}(u,y)}=0$ almost surely, and thus $\varphi(u,y)$ is unique. Please refer to Appendix A.1 for more detailed proof. This is a necessary condition for nonparametric identification of $\phi(u,y)$. However, when $\phi(u,y)$ belongs to a parametric/semiparametric model class, the completeness condition can be weakened. For example, Shao and Wang (2016) considered a semiparametric model for $\phi(u,y)$ and establish identification with relaxed assumptions. We will include detailed discussion in the revised version.
>
> Reference: Shao, J. and Wang, L., 2016. Semiparametric inverse propensity weighting for nonignorable missing data. Biometrika, 103(1), pp.175-187.
>
> > 2. Regarding assumption 4.2, it is worthwhile to discuss in the paper how restrictive this assumption is. Perhaps the authors could mention some cases where it does not hold.
>
> **Re**: Thanks for the constructive comment. We will include a more detailed discussion by incorporating our responses to the weaknesses and Question 1 in the revised version. We believe this will offer a clearer and more straightforward interpretation of this assumption.
>
> > 3. How do high-dimensional categorical covariates affect the estimation for the conditional expectations as required by assumption 5.1?
>
> **Re**: Please refer to our response to the weaknesses.
>
> > 4. The authors mentioned that the maximizer for the efficient estimator denoted by $\hat{\pi}$ is a natural estimator for the true optimal decision rule $\pi^*$, which is a maximizer of the true value function. Although in experiments, the authors show percentages of making correct decisions (PCD), can they provide any discussion on how far $\hat{\pi}$ can be from the true optimal rule $\pi^*$ in general?
>
> **Re**: Thanks for the insightful question. Generally, the convergence rate of $\hat{\pi}$ to $\pi^*$ depends on the complexity of the decision rule class $\Pi$. For smooth parametric decision rules, $\hat{\pi}$ converges to $\pi^*$ at a rate of $n^{-1/2}$ (Wu and Wang, 2021). In contrast, for non-smooth parametric decision rules—such as $\mathbb{I}\\{\beta^T\theta(X)>0\\}$, where $\theta(X)$ represents a specific transformation of the covariates—the convergence rate is $n^{-1/3}$ (Wang et al. 2018). For nonparametric decision rules, the rate of convergence depends on the complexity of the decision rule class. We will include a detailed discussion of these points in the revised version.
>
> Reference:
>
> Wu, Y. and Wang, L., 2021. Resampling-based confidence intervals for model-free robust inference on optimal treatment regimes. Biometrics, 77(2), pp.465-476.
>
> Wang, L., Zhou, Y., Song, R. and Sherwood, B., 2018. Quantile-optimal treatment regimes. Journal of the American Statistical Association, 113(523), pp.1243-1254.

---

> > ### Author Response · Authors · 2024-11-22
> >
> > We are sincerely grateful for the time and effort you have devoted to providing detailed, insightful, and encouraging comments on our submission. In response to your constructive questions, we have updated our discussions accordingly. We are eagerly looking forward to your feedback and would be happy to make further adjustments if there are additional changes you would like to suggest.
> >
> > Your encouraging remarks and valuable recommendations are greatly appreciated, and we thank you for your support throughout the review process!

---

> > > ### Comment · Reviewer_Wzn4 · 2024-11-23
> > > **Re.**
> > >
> > > Thanks to the authors for their detailed response. The clarity of the paper will be improved by including discussions regarding the assumptions. I will keep my score the same, and I am inclined toward the acceptance decision for the  paper.

---

### Meta-Review · Area_Chair_Ft9g · 2024-12-22

**Metareview:**

Although one-sided information appears in contexts such as classification, the paper takes the unexplored path of looking into one-sided bandit feedback, and which kind of indirect information can be used in this case, along with explicit assumptions and their real-world appeal. There is a relationship with selection bias problems and many applications found in the causal inference literature. The paper explorers those links, including the ideas of efficient influence function approaches for estimation. All reviewers agreed this is a valuable contribution, and provided feedback about the plausibility of assumptions and doable ways of improving clarity.

**Additional Comments On Reviewer Discussion:**

All reviewers agreed this is a valuable contribution, and provided feedback about the plausibility of assumptions and doable ways of improving clarity.

---

### Decision · Program_Chairs · 2025-01-22

Accept (Poster)